# Implicit Reasoning in Large Language Models: A Comprehensive Survey

## Abstract

Large Language Models (LLMs) have demonstrated strong generalization across a wide range of tasks. Reasoning with LLMs is central to solving multi-step problems and complex decision-making. To support efficient reasoning, recent studies have shifted attention from explicit chain-of-thought prompting toward implicit reasoning, where reasoning occurs silently via latent structures without emitting intermediate textual steps. Implicit reasoning brings advantages such as lower generation cost, faster inference, and better alignment with internal computation. Although prior surveys have discussed latent representations in the context of reasoning, a dedicated and mechanism-level examination of how reasoning unfolds internally within LLMs remains absent. This survey fills that gap by introducing a taxonomy centered on execution paradigms, shifting the focus from representational forms to computational strategies. We organize existing methods into three execution paradigms based on ***how and where internal computation unfolds***: latent optimization, signal-guided control, and layer-recurrent execution. We also review structural, behavioral and representation-based evidence that supports the presence of implicit reasoning in LLMs. We further provide a structured overview of the evaluation metrics and benchmarks used in existing works to assess the effectiveness and reliability of implicit reasoning. We maintain a continuously updated project at: https://anonymous.4open.science/r/TMLR-LLM-Implicit-Reasoning-Survey-E4D6.

## 1 Introduction

In recent years, large language models (LLMs) (Touvron et al., 2023a; Li et al., 2023b; Javaheripi et al., 2023; Abdin et al., 2024; Grattafiori et al., 2024; Hurst et al., 2024; Contributors et al., 2024; Yang et al., 2024a;b; 2025a; Guo et al., 2025; OpenAI, 2025; DeepSeek-AI, 2025) have made significant advances in a broad spectrum of tasks (Liu et al., 2025a), including but not limited to dialogue generation (Yi et al., 2024), recommender systems (Wang et al., 2024b), healthcare (Qiu et al., 2024), finance (Li et al., 2023a), test-time compute (Li, 2025), tabular data (Fang et al., 2025b), and scientific reasoning (Yan et al., 2025) thanks to their large number of parameters and massive training data. However, research shows that simply relying on linear growth in parameter count is not enough to explain all performance gains. During inference, test-time scaling (TTS) (Zhang et al., 2025b) reveals the model's capability for "dynamic computation", that is, investing extra computing resources at inference time to achieve a deeper understanding and reasoning. Typical examples of this idea are *o1* (Contributors et al., 2024) and *DeepSeek R1* (Guo et al., 2025), which are reasoning models that gained strong performance.

Most recent reasoning models depend on explicit Chain-of-Thought (CoT) (Wei et al., 2022), where the models first "say out" a coherent series of intermediate reasoning steps in natural language (Sun et al., 2023) and then give the final answer, thus significantly improving accuracy on complex problems. Although explicit reasoning can improve interpretability and depth, it can often lead to much longer sequences because of lengthy, unnecessary, or irrelevant steps (Hong et al., 2025), which waste computing resources and increase latency and cost in real applications (Yue et al., 2025a), as shown in Figure 1. To address this, the research community has started to explore new ways to keep deep reasoning ability while improving reasoning efficiency and reducing the burden of "overthinking" (Sui et al., 2025).

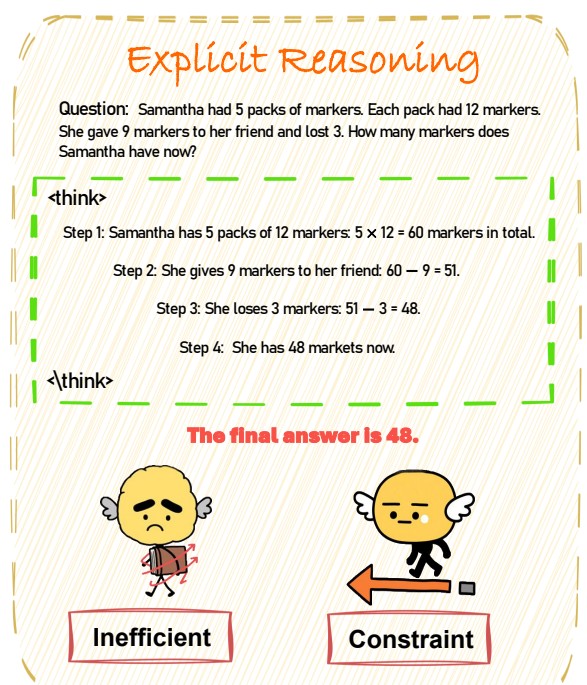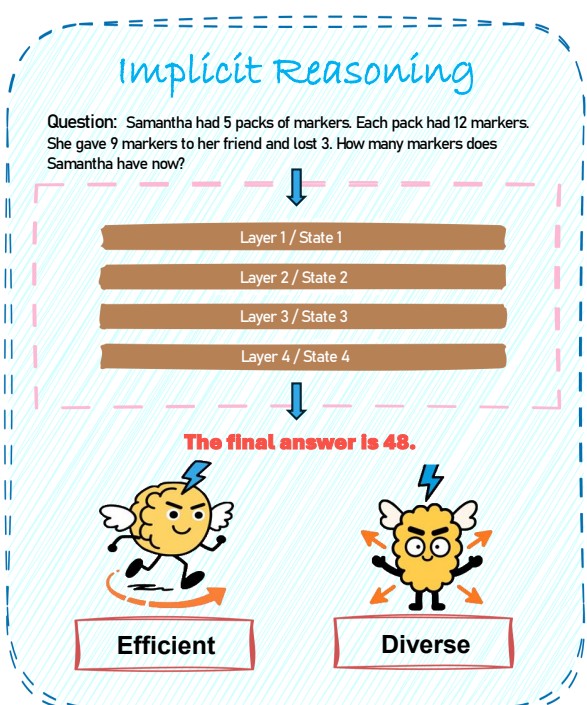

Figure 1: Comparison between explicit and implicit reasoning in LLMs. Explicit reasoning shows each step by producing natural language explanations, as illustrated on the left. The model describes the problem-solving process one step at a time. In contrast, implicit reasoning, shown on the right, handles the process internally across different layers or states without writing out any steps. Explicit reasoning is less efficient because generating text takes time and resources. *Implicit reasoning happens inside the model by hidden representations, supporting faster processing.* Also, explicit reasoning is limited by the structure of language, while *implicit reasoning allows many types of internal computation without needing to be described in words.*

To this end, recent studies have introduced the concept of implicit reasoning (Ye et al., 2025a), where multi-step reasoning is performed without emitting explicit reasoning traces. Rather than producing visible intermediate steps, the model carries out reasoning internally via token-level (Tack et al., 2025; Sun et al., 2025), trajectory-level (Cheng & Van Durme, 2024; Hao et al., 2024), internal-state-level latent refinement (Deng et al., 2023; Kong et al., 2025) or signal-guided control (Herel & Mikolov, 2024; Goyal et al., 2024; Pfau et al., 2024; Wang et al., 2024c), etc. This silent form of reasoning reduces surface complexity and may better align with how reasoning unfolds inside the model. Despite increasing attention, implicit reasoning remains underexplored and calls for a more systematic understanding.

LLM implicit reasoning breaks free from the need to output tokens at each step of reasoning, and completes the process directly in the model's continuous representation space. This method does not require converting each reasoning step into natural language tokens, as shown in Figure 1, so it avoids the computational and serialization bottleneck of multiple autoregressive generations and can run reasoning in parallel inside the model more efficiently. By using more efficient internal structures, such as latent embeddings and neural network layers, implicit reasoning not only makes better use of resources (Hao et al., 2024; Zhang et al., 2025a) but also can explore more diverse reasoning paths (Xu et al., 2025a; Gozeten et al., 2025) without the constraints of decoding.

Despite growing interest in implicit reasoning, the literature remains fragmented. Existing works span multiple directions, including latent-state modeling, compact reasoning trajectories, loop-based computation, and test-time control, yet lack a unified conceptual framework. Though several prior surveys have reviewed LLM reasoning more broadly (Ahn et al., 2024; Zhou et al., 2025; Chen et al., 2025a; Li et al., 2025b), these mostly focus on explicit paradigms (Qu et al., 2025; Liu et al., 2025b; Feng et al., 2025; Wang et al., 2025b;

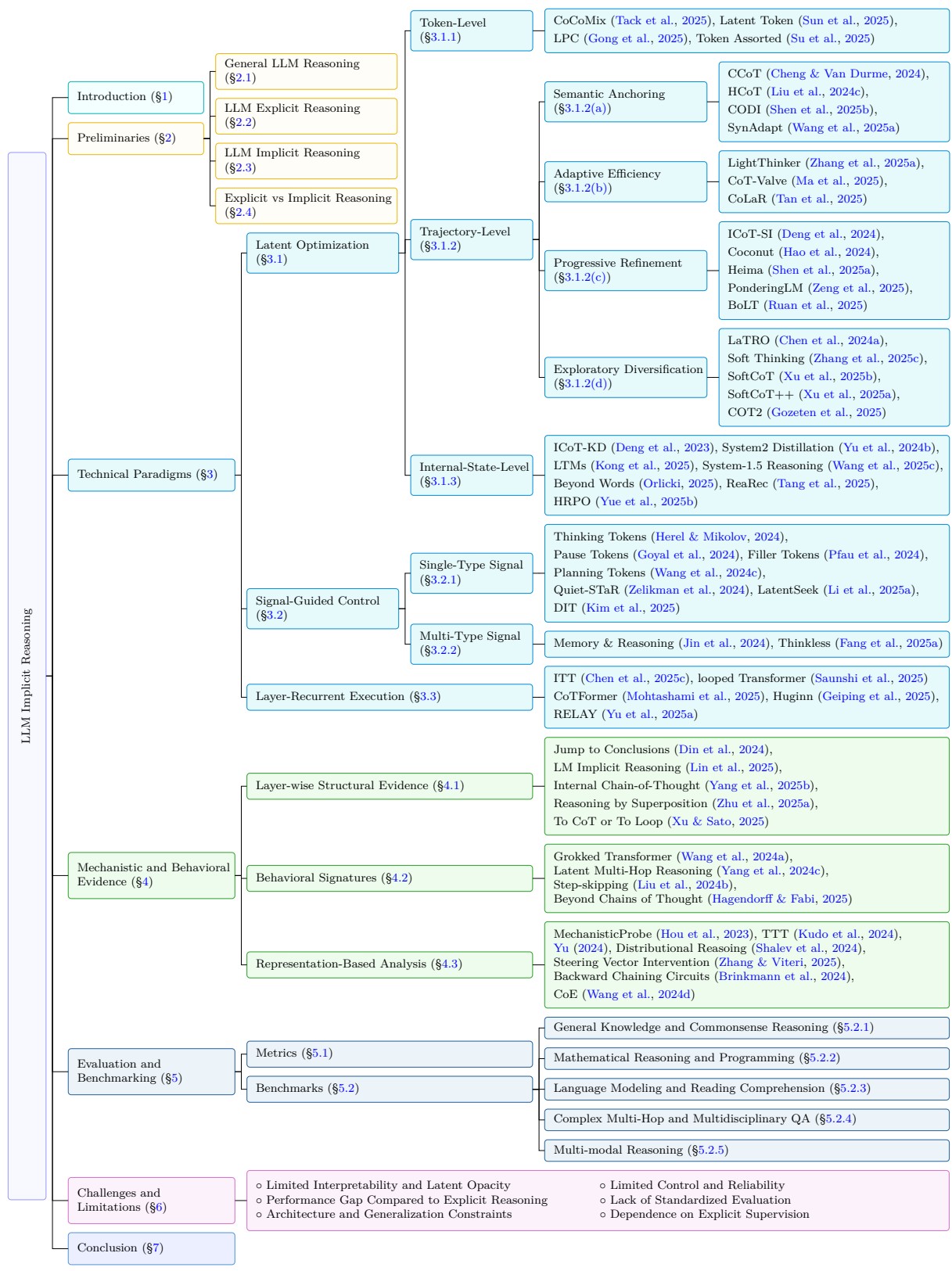

Figure 2: Taxonomy of this paper with representative works.

Sui et al., 2025) such as CoT prompting or symbolic reasoning, leaving implicit reasoning underexplored. A few recent surveys have touched upon latent forms of reasoning (Chen et al., 2025b; Zhu et al., 2025b), yet their scopes differ substantially from ours. Specifically, Chen et al. (2025b) structure the field from four perspectives: token-wise strategies, internal mechanisms, analysis, and applications, emphasizing how Chain-of-Thought reasoning can be re-encoded into latent forms. Zhu et al. (2025b) take a mechanistic viewpoint, focusing on architectural recurrence, temporal hidden states, and layer-wise interpretability.

To consolidate the fragmented literature and clarify this emerging paradigm, we present a systematic survey of implicit reasoning in LLMs from a functional perspective. We organize existing methods according to ***how and where internal computation unfolds***, forming a taxonomy comprising three execution paradigms (§3): latent optimization (§3.1), signal-guided control (§3.2), and layer-recurrent execution (§3.3). In addition to categorizing methods, we analyze the structural, behavioral, and representation-based evidence that supports the presence of implicit reasoning (§4). We also provide a structured overview of evaluation metrics and benchmarks commonly adopted across the literature (§5), an aspect largely overlooked in prior surveys. By establishing a coherent framework, this survey aims to unify diverse efforts and support future research toward efficient, controllable, and cognitively grounded reasoning, while also identifying key challenges and outlining promising future directions (§6). The overall structure of our survey is illustrated in Figure 2.

Our contribution can be summarized as follows:

- To systematically characterize implicit reasoning in LLMs, we introduce a functional perspective that emphasizes ***how and where internal computation unfolds***. Based on this view, we establish an execution-centric taxonomy comprising three paradigms: latent optimization, signal-guided control, and layer-recurrent execution, each further refined into subtypes according to reasoning granularity and control mechanisms.

- We conduct a parallel investigation into the evidence for implicit reasoning by synthesizing findings from structural analyses, behavioral signatures, and representation-based analysis techniques, providing empirical grounding for the internal dynamics captured by our execution-centric taxonomy.

- We conduct a systematic review of evaluation protocols and benchmarking practices commonly adopted in the study of implicit reasoning. We also identify pressing challenges in advancing the field and outline future directions for building reasoning systems that are more efficient, robust, interpretable, and cognitively aligned.

## 2 Preliminaries

This section establishes key notations and definitions for reasoning in large language models (LLMs). We formally distinguish between *explicit reasoning* and *implicit reasoning*, and describe their respective characteristics from the perspective of execution and representation.

### 2.1 General LLM Reasoning

Large Language Models (LLMs) like the GPT (Hurst et al., 2024; OpenAI, 2025), DeepSeek (Liu et al., 2024a; Guo et al., 2025; DeepSeek-AI, 2025) and Qwen (Yang et al., 2024a;b; 2025a; Team, 2025) families, excel on tasks that require more-than-one-step prediction, including commonsense QA (Talmor et al., 2019), mathematical reasoning (Cobbe et al., 2021; Hendrycks et al., 2021b), multi-hop QA (Yang et al., 2018), and multi-modal reasoning (Chen et al., 2024b). Unlike static classification, these tasks demand a sequence of intermediate computations before arriving at the correct final answer.

We formalize **LLM reasoning** as a two-stage inference process carried out by a model $\pi_\theta$ given an input $x$. In the first stage, the model generates an internal trace $z_{1:M}$, where

$$z_{1:M} = (z_1, \ldots, z_M) \tag{1}$$

is the sequence of $M$ intermediate reasoning steps. Each $z_t$ may be a sequence of natural-language tokens (Wei et al., 2022), a hidden state (Hao et al., 2024), or the output of an internal layer (Saunshi et al., 2025). In the second stage, the model emits the final answer $a$ conditioned on $x$ and the trace $z_{1:M}$.

In a simplified form, the two steps can be written as

$$z_{1:M} \sim \pi_\theta(\cdot \mid x),$$
$$a \sim \pi_\theta(\cdot \mid x, z_{1:M}). \tag{2}$$

This decomposition shows how the model first builds an internal reasoning trace and then uses it to produce the answer. When the steps $z_{1:M}$ are itself emitted as text alongside $a$, we call the process *explicit reasoning*. When only $a$ is produced and $z_{1:M}$ remains internal, we call it *implicit reasoning* (Chen et al., 2025b; Zhu et al., 2025b). Both follow the same two-stage formulation, differing apparently in whether the trace is visible to the user.

## 2.2 LLM Explicit Reasoning

When the model is guided or trained to show each intermediate reasoning step in natural language alongside the final answer (Wei et al., 2022; Chen et al., 2025a), we call the process *explicit reasoning*.

**Definition 1** (Explicit Reasoning). *We define **explicit reasoning** as the paradigm in which the model first generates a sequence of textual reasoning steps*

$$y_{1:T} \sim \pi_\theta(\cdot \mid x), \tag{3}$$

*where each $y_t$ in $y_{1:T}$ is the natural-language form of the t-th reasoning step, and then emits the final answer*

$$a \sim \pi_\theta(\cdot \mid x, y_{1:T}). \tag{4}$$

*This formulation is a simplified notation; in practice, each step $y_t$ is generated autoregressively conditioned on $x$ and the previous steps $y_{1:t-1}$.*

## 2.3 LLM Implicit Reasoning

In contrast, *implicit reasoning* refers to settings where the model performs multi-step inference internally without generating any intermediate steps as output (Chen et al., 2025b; Zhu et al., 2025b). The reasoning unfolds implicitly through multiple paradigms, including latent optimization (token-level (§3.1.1), trajectory-level (§3.1.2), internal-state-level (§3.1.3)), signal-guided control (single-type signal (§3.2.1), multi-type signal (§3.2.2)), and layer-recurrent execution (§3.3), with only the final output exposed.

**Definition 2** (Implicit Reasoning). *We define **implicit reasoning** as the paradigm in which the model first generates a hidden trace*

$$h_{1:L} \sim \pi_\theta(\cdot \mid x), \tag{5}$$

*where $h_{1:L}$ is a sequence of L internal states (e.g. hidden activations or latent tokens), and then emits the final answer*

$$a \sim \pi_\theta(\cdot \mid x, h_{1:L}). \tag{6}$$

*This formulation is a simplified notation; although the hidden states $h_{1:L}$ are generally generated autoregressively, they remain entirely internal and invisible to the model's output.*

## 2.4 Explicit vs. Implicit Reasoning

Explicit and implicit reasoning diverge in how reasoning is structured, executed, and interpreted (Chen et al., 2025b). Their differences span multiple dimensions, including visibility, supervision, efficiency, interpretability, alignment with human thinking, and diversity of reasoning trajectories. We detail these dimensions in the following subsections.

**Reasoning Visibility.** Explicit reasoning verbalizes intermediate reasoning states in natural language, producing interpretable chains such as "Step 1: ... Step 2: ..." (Wei et al., 2022). This makes the reasoning process transparent and easy to inspect. In contrast, implicit reasoning suppresses intermediate traces (Cheng & Van Durme, 2024; Hao et al., 2024), with all multi-step computation absorbed into the model's internal hidden states, attention patterns, or latent variables that are not directly accessible.

Table 1: Key differences between explicit reasoning and implicit reasoning in LLMs. (§2.4)

| Dimension | Explicit Reasoning | Implicit Reasoning |
|---|---|---|
| **Reasoning Visibility** | States verbalized in text, transparent | States hidden in latent space, invisible |
| **Reasoning Efficiency** | Verbose, high cost and latency | Compact, faster, resource-efficient |
| **Interpretability** | Directly observable and checkable | Indirect, via probing or attribution |
| **Reasoning Diversity** | Commits to one trajectory | Encodes multiple alternatives |
| **Supervision Granularity** | Explicit, step-aware supervision | Guided by latent objectives |
| **Alignment with Human Thinking** | Explains thoughts aloud | Thinking silently |

**Reasoning Efficiency.**   Explicit reasoning easily suffers from low efficiency due to verbose natural-language outputs of each steps, leading to increased decoding cost and latency (Cheng & Van Durme, 2024; Liu et al., 2024c; Shen et al., 2025a). This overhead is particularly pronounced for complex tasks. Instead, implicit reasoning avoids verbose token generation and achieves faster reasoning with reduced resource consumption (Tan et al., 2025; Chen et al., 2025b).

**Interpretability.**   Explicit reasoning is easy to interpret, as the full reasoning path is observable and can be manually assessed for logical consistency. In contrast, implicit reasoning is hidden, and understanding it requires indirect analysis: researchers may probe hidden states (Yu, 2024), visualize attention flows (Lin et al., 2025), or analyze prediction behaviors (Wang et al., 2024a) to infer whether meaningful reasoning occurred.

**Reasoning Diversity.**   Explicit reasoning need to sample tokens form a finite vocabulary and verbalize intermediate reasoning steps in fixed semantic space, easily committing to one specific reasoning trajectory and lack of possible reasoning exploration (Zhang et al., 2025c; Gozeten et al., 2025). In contrast, implicit reasoning is silently performed and can encode multiple alternative reasoning trajectories in latent space, naturally exploring richer diversity (Xu et al., 2025a).

**Supervision Granularity.**   Explicit reasoning easily allows prompt-level guidance (Wei et al., 2022) or loss-level supervision over each reasoning step, enabling human steering and fine-tuning. In contrast, implicit reasoning has less direct supervision; the internal reasoning is shaped via latent objectives or emergent behaviors during training (Liu et al., 2024c; Xu et al., 2025a; Wang et al., 2024a).

**Alignment with Human Thinking.**   Implicit reasoning arguably resembles how humans think silently, performing mental computation and only outputting the final answer, while explicit reasoning mimics how humans explain their thoughts aloud (Yu et al., 2024b; Wang et al., 2025c; Orlicki, 2025). Both are cognitively relevant, but support different use cases and evaluation protocols.

These distinctions between explicit and implicit reasoning motivate different research directions. While explicit reasoning supports interpretability and supervision, it can be verbose and inefficient. Implicit reasoning, in contrast, is efficient and compact but less transparent, raising unique challenges for analysis and evaluation.

## 3 Technical Paradigms for Implicit Reasoning

To systematize existing efforts in modeling implicit reasoning, we categorize current methods into three complementary paradigms based on where and how latent reasoning is formed within the model. The first paradigm, *latent optimization* (§3.1), directly manipulates internal representations to improve reasoning without emitting intermediate text. The second, *signal-guided control* (§3.2), leverages specially designed control signals to steer the model's internal computation process. The third, *layer-recurrent execution* (§3.3), introduces iterative computation within the model's architecture to progressively refine hidden states. These

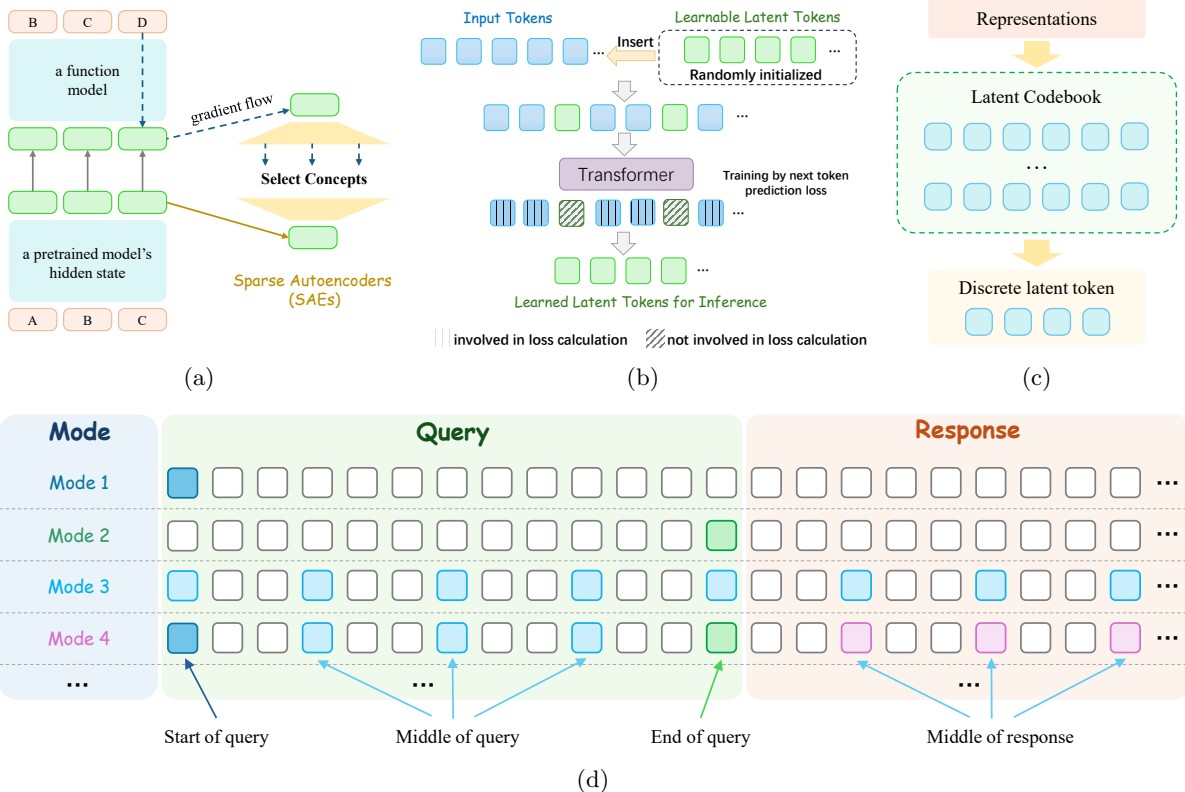

Figure 3: Token-level latent optimization. Illustration of representative paradigms among diverse strategies for acquiring and utilizing special latent tokens: (a) Concept tokens selected from pretrained hidden states via sparse autoencoders (Tack et al., 2025). (b) Learnable latent tokens optimized by a next token prediction loss (Sun et al., 2025). (c) Discrete latent tokens via vector quantization (Gong et al., 2025; Su et al., 2025). (d) Common usage patterns of latent representation tokens, illustrating how they are interleaved with standard tokens at different positions (e.g., start/middle of query or response) (Sun et al., 2025).

paradigms reflect distinct yet compatible strategies for enhancing the internal reasoning abilities of LLMs, and structure the technical survey that follows.

### 3.1 Latent Optimization for Implicit Reasoning

Latent optimization methods improve reasoning by directly adjusting and optimizing internal representations without emitting intermediate text, allowing models to internalize reasoning as a continuous process over latent units. Depending on the granularity of the optimized target unit, existing approaches can be grouped into three types: *token*-level (§3.1.1), *trajectory*-level (§3.1.2), and *internal-state*-level (§3.1.3). This taxonomy reflects distinct ways of localizing and manipulating reasoning within the model's latent space.

#### 3.1.1 Token-Level

Token-level latent optimization methods (see Table 2) steer reasoning by manipulating **individual tokens**. They may insert semantic concepts (Tack et al., 2025) or non-interpretable latent tokens (Sun et al., 2025) into reasoning steps, learn discrete latent codes to guide preference-aware generation (Gong et al., 2025), or replace spans of text with compact latent tokens for compressed reasoning (Su et al., 2025), as illustrated in Figure 3.

Concretely, *CoCoMix* (Tack et al., 2025) extracts continuous semantic concepts from a pretrained sparse autoencoder (SAE) (Cunningham et al., 2023), and integrates them into the language model's hidden states

Table 2: Token-level latent optimization (§3.1.1).

| Models | Main Techniques | Base Model | Main Tasks, Scenarios | Main Datasets | Open Source |
|---|---|---|---|---|---|
| **CoCoMix** (Tack et al., 2025) | Sparse autoencoders (SAEs), continuous concept mixing | GPT-2 (Radford et al., 2019) | Commonsense reasoning, reading comprehension, weak-to-strong supervision | LAMBADA (Paperno et al., 2016), WikiText-103 (Merity et al., 2017), HellaSwag (Zellers et al., 2019), PIQA (Bisk et al., 2020), Social IQA (Sap et al., 2019), ARC-easy (Clark et al., 2018), WinoGrande (Sakaguchi et al., 2020); OpenWeb-Math (Paster et al., 2024) | GitHub |
| **Latent Token** (Sun et al., 2025) | Inference with latent tokens, position encoding of latent tokens, design choices | LLaMA-3.1-8B (Grattafiori et al., 2024), LLaMA-3.2-1B (Meta, 2024) | Language modeling, Reading comprehension, arithmetic reasoning | WikiSplit (Botha et al., 2018), NarrativeQA (Kočiský et al., 2018), GSM8K (Cobbe et al., 2021) | - |
| **LPC** (Gong et al., 2025) | Discrete latent codes, a prior network and a posterior network, Reinforcement Learning from Human Feedback (RLHF) | Mistral-7B (Jiang et al., 2023), LLaMA-3-8B (Grattafiori et al., 2024), LLaMA-3-8B-Instruct (Grattafiori et al., 2024) | Common reasoning, mathematical reasoning, truthfulness | TruthfulQA (Lin et al., 2022), ARC-easy (Clark et al., 2018), ARC-challenge (Clark et al., 2018), GSM8K (Cobbe et al., 2021); UltraFeedback (Cui et al., 2023) | - |
| **Token Assorted** (Su et al., 2025) | Latent discrete token, latent trace abstraction | T5 (Raffel et al., 2020), GPT-2 (Radford et al., 2019), LLaMA-3.1-8B (Grattafiori et al., 2024), LLaMA-3.2-1B (Meta, 2024), LLaMA-3.2-3B (Meta, 2024) | Multi-step planning, logical reasoning, mathematical reasoning | Keys-Finding Maze (Su et al., 2025), ProntoQA (Saparov & He, 2023), ProsQA (Hao et al., 2024), MATH (Hendrycks et al., 2021b), GSM8K (Cobbe et al., 2021), College-Math (Tang et al., 2024), Mathematics Dataset (Saxton et al., 2019), OlympiadBench-Math (He et al., 2024), TheoremQA (Chen et al., 2023), Fresh-Gaokao-Math-2023 (Tang et al., 2024); MetaMathQA (Yu et al., 2024a), Dart-MATH (Tong et al., 2024) | - |

to enhance next-token prediction. By selecting salient concepts via attribution scores and interleaving their compressed forms with token representations, *CoCoMix* bridges surface-level tokens with high-level semantics, enabling improved reasoning, interpretability, and controllable generation. *Latent Token* (Sun et al., 2025) enhances reasoning ability and generalization to out-of-distribution scenarios by inserting non-interpretable tokens into Transformer inputs, which can be flexibly placed at arbitrary positions within the sequence to enable fine-grained control over the computation process, all without modifying the backbone model. *Latent Preference Coding (LPC)* (Gong et al., 2025) employs discrete latent codes to model implicit factors and their combinations behind holistic preferences without predefined rewards or hand-crafted weights, guiding preference-aware generation of LLMs, such as rigorous reasoning needed in mathematical tasks. *Token Assorted* (Su et al., 2025) introduces a hybrid reasoning format by interleaving discrete latent tokens abstracted by VQ-VAE with text tokens to compress reasoning processes. The model is trained with a simple mixing strategy and an extended vocabulary, enabling fast adaptation to latent abstractions and improved performance on logical and mathematical reasoning tasks.

### 3.1.2 Trajectory-Level

Unlike token-level approaches that adjust individual tokens, trajectory-level methods treat the reasoning trajectory as a unit of optimization, replacing explicit reasoning steps with continuous latent thoughts. Specifically, these methods typically intervene at the granularity of **reasoning steps** and compress explicit reasoning steps into compact latent trajectories, which are anchored to explicit reasoning semantically, ensuring semantic fidelity while reducing decoding overhead (§3.1.2(a)). Beyond this, some research further develops the paradigm by introducing dynamically adaptive mechanisms (§3.1.2(b)), progressive refinement (§3.1.2(c)), and exploratory diversification of multiple latent trajectories (§3.1.2(d)). Representative designs are illustrated in Figure 4, and key statistics are summarized in Table 3.

**(a) Semantic Anchoring.** Within trajectory-level methods, the most fundamental approach is to directly *anchor* latent trajectories to explicit reasoning supervision (Cheng & Van Durme, 2024). This paradigm can be viewed as the default mechanism underlying trajectory-level methods: latent trajectories are compressed from multi-step reasoning traces and guided to preserve their essential semantics faithfully. Although conceptually simple, this strategy establishes semantic fidelity as a foundation of trajectory-level optimization, and serving as the basis upon which more adaptive or exploratory techniques are developed.

Specifically, *Compressed Chain of Thought (CCoT)* (Cheng & Van Durme, 2024) compresses full reasoning traces into contentful and continuous contemplation tokens in latent space. Specifically, CCoT employs a scorer module to select the subset of gold hidden states and generates compressed tokens to approximate

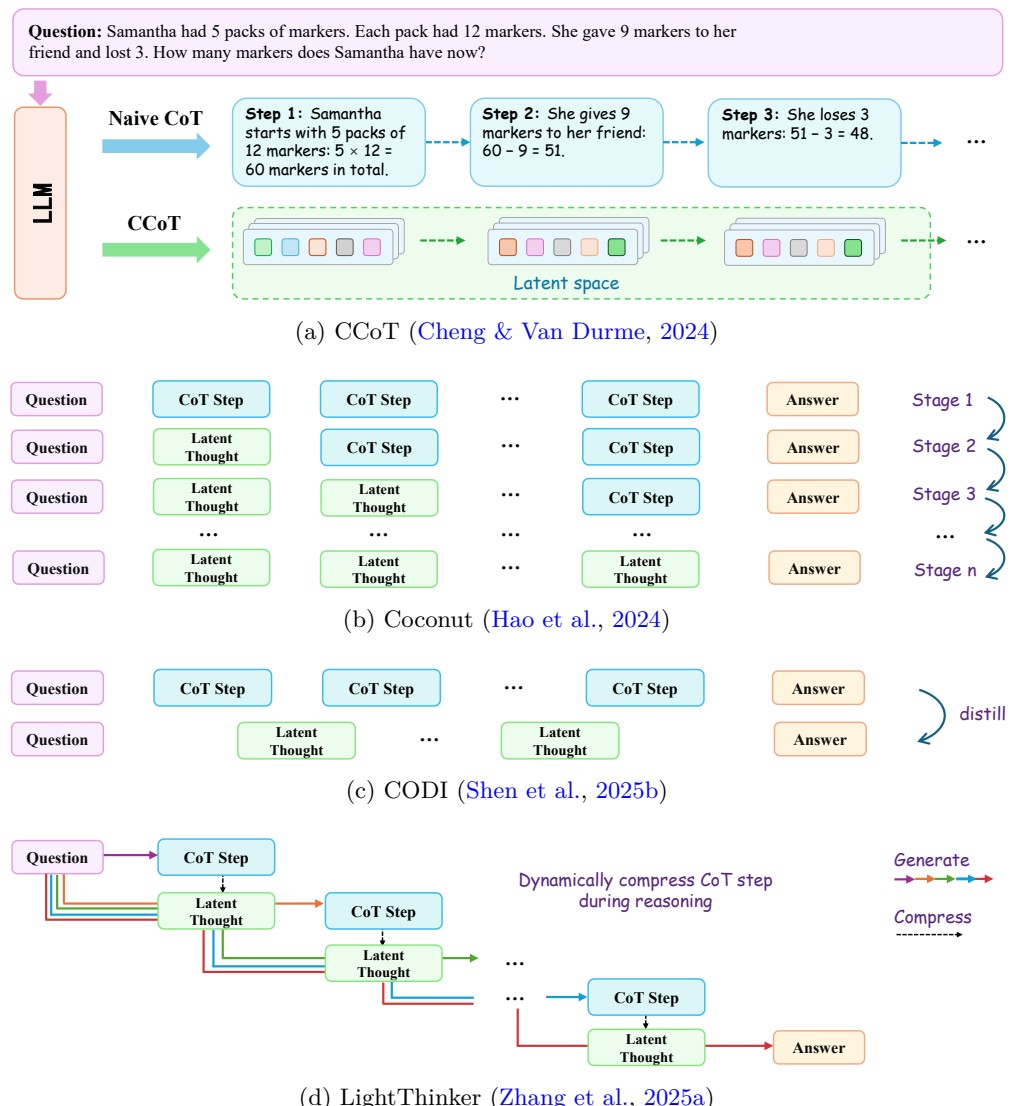

Figure 4: Trajectory-level latent optimization. Illustration of representative methods for encoding multi-step reasoning trajectories in latent space: (a) *CCoT* compresses the full CoT traces into short sequences of continuous embeddings, reducing decoding cost while preserving essential reasoning semantics (Cheng & Van Durme, 2024). (b) *Coconut* replaces discrete reasoning steps with latent thoughts in a multi-stage training process, enabling latent reasoning progressively (Hao et al., 2024). (c) *CODI* distills explicit CoTs into continuous latent thoughts under a self-distillation framework in a single-stage compression manner (Shen et al., 2025b). (d) *LightThinker* (Zhang et al., 2025a) dynamically compresses reasoning steps into latent gist tokens at the designated position, reducing memory and computational overhead.

and align these subsets, supporting reduced decoding cost and seamless integration into pretrained decoder-only LLMs via lightweight finetuning. *Hidden Chain-of-Thought (HCoT)* (Liu et al., 2024c) compresses the full reasoning traces into a special [*CoT*] token, semantically aligned through contrastive training with an auxiliary CoT model, and then predicts final answers based on these aligned tokens. By disentangling the training of the auxiliary model and the downstream predictor, HCoT enables modular optimization and interpretable reasoning compression. To avoid forgetting issues in curriculum learning, *CODI* (Shen et al., 2025b) establishes a self-distillation framework that aligns hidden states at a key token of answer generation between explicit and implicit CoT tasks, effectively compressing reasoning into continuous space. However, these methods often employ a single reasoning token or the subset of reasoning tokens for semantic anchor-

Table 3: Trajectory-level latent optimization (§3.1.2).

| Models | Main Techniques | Base Model | Main Tasks, Scenarios | Main Datasets | Open Source |
|---|---|---|---|---|---|
| **CCoT** (Cheng & Van Durme, 2024) | Contemplation tokens, compressed representations of language-based reasoning chains | LLaMA-2-7B-Chat (Touvron et al., 2023b) | Mathematical reasoning | GSM8K (Cobbe et al., 2021) | **-** |
| **HCoT** (Liu et al., 2024c) | Two stage, disentangled training paradigm, compress CoT process, compact special token, contrastive learning | LLaMA-2-7B (Touvron et al., 2023b), LLaMA-2-13B (Touvron et al., 2023b) | Mathematical reasoning, agent invocation, science question answering | GSM8K (Cobbe et al., 2021), MATH (Hendrycks et al., 2021b), ScienceQA (Lu et al., 2022), HotpotQA (Yang et al., 2018) | **-** |
| **CODI** (Shen et al., 2025b) | Compress CoT into continuous space, self distillation | GPT-2 (Radford et al., 2019), LLaMA-3.2-1B-Instruct (Meta, 2024) | Mathematical reasoning, compress more verbose CoTs, Commonsense reasoning, out-of-distribution (OOD) evaluation | GSM8K (Cobbe et al., 2021), SVAMP (Patel et al., 2021), GSM-Hard (Gao et al., 2023), Multi-Arith (Roy & Roth, 2015), CommonsenseQA (Talmor et al., 2019) | GitHub |
| **SynAdapt** (War et al., 2025a) | Adaptive reasoning, synthetic continuous CoT, comprehensive alignment, accuracy-efficiency trade-off | DeepSeek-R1-Distill-Qwen-7B (Guo et al., 2025), DeepSeek-R1-Distill-LLaMA-8B (Guo et al., 2025), DeepSeek-R1-Distill-Qwen-1.5B (Guo et al., 2025) | Mathematical reasoning | GSM8K (Cobbe et al., 2021), MATH-500 (Lightman et al., 2024), AMC23 (zwhe99, 2024), AIME24 (MathAI, 2024a), AIME25 (MathAI, 2024b) | - |
| **LightThinker** (Zhang et al., 2025a) | Dynamically compresses intermediate thoughts during generation, data reconstruction, thought-based attention mask construction | Qwen2.5-7B series (Yang et al., 2024b), LLaMA-3.1-8B series (Grattafiori et al., 2024), DeepSeek-R1-Distill (Guo et al., 2025) | Mathematical reasoning, logical reasoning | GSM8K Cobbe et al. (2021), MMLU (Hendrycks et al., 2021a), GPQA (Rein et al., 2024), BIG-Bench Hard (BBH) (Suzgun et al., 2023); Bespoke-Stratos-17k (BS17K) (Bespoke Labs, 2025) | GitHub |
| **CoT-Valve** (Ma et al., 2025) | Length-compressible CoT tuning | LLaMA-3.1-8B (Grattafiori et al., 2024), LLaMA-3.2-1.5B-Instruct (Meta, 2024), QwQ-32B-Preivew (Team, 2024), DeepSeek-R1-Distill-LLaMA-8B (Guo et al., 2025), Qwen2.5-32B-Instruct (Yang et al., 2024b) with LIMO (Ye et al., 2025b) | Long to short CoT, short to long CoT, short-long-short CoT | GSM8K (Cobbe et al., 2021), AIME24 (MathAI, 2024a), MixChain (Ma et al., 2025) | GitHub |
| **CoLaR** (Tan et al., 2025) | Performs reasoning at a dense latent level (silently), dynamically adjusts reasoning speed, GRPO (Shao et al., 2024; Yu et al., 2025b) | LLaMA-3.2-1B-Instruct (Grattafiori et al., 2024) | Mathematical reasoning | GSM8K (Cobbe et al., 2021), GSM8K-Hard (Gao et al., 2023), SVAMP (Patel et al., 2021), Multi-Arith (Roy & Roth, 2015) | GitHub, HomePage |
| **ICoT-SI** (Deng et al., 2024) | Curriculum learning, stepwise internalization | GPT-2 Small (Radford et al., 2019), GPT-2 Medium (Radford et al., 2019), Phi-3-3.8B (Abdin et al., 2024), Mistral-7B (Jiang et al., 2023) | Multi-digit multiplication, Grade school math problem | BIG-Bench (Srivastava et al., 2023), GSM8K-Aug (Deng et al., 2023) | GitHub |
| **Coconut** (Hao et al., 2024) | Continuous thought, unrestricted latent space, encode multiple potential next steps simultaneously | GPT-2 (Radford et al., 2019) | Math reasoning, planning-intense tasks | GSM8K (Cobbe et al., 2021), ProsQA (Hao et al., 2024), ProntoQA (Saparov & He, 2023) | GitHub |
| **Heima** (Shen et al., 2025a) | Thinking token, CoT reconstruction, multimodal | LLaVA-CoT (Xu et al., 2024), LLaMA-3.1-8B-Instruct (Grattafiori et al., 2024) | Multimodal reasoning | LLaVA-CoT-100K (Xu et al., 2024), MMStar (Chen et al., 2024b), MMBench (Liu et al., 2024d), MM-Vet (Yu et al., 2024c), MathVista (Lu et al., 2024), AI2D-RST (Hiippala et al., 2021), Hallusion-Bench (Guan et al., 2024) | GitHub |
| **PonderingLM** (Zeng et al., 2025) | Pondering, produces a weighted sum of all token embeddings based on the predicted probabilities, self-supervised learning | GPT-2 (Radford et al., 2019), Pythia (Biderman et al., 2023), LLaMA (Meta AI, 2023) architectures | Commonsense reasoning, reading comprehension | LAMBADA (Paperno et al., 2016), PIQA (Bisk et al., 2020), WinoGrande (Sakaguchi et al., 2020), ARC-easy (Clark et al., 2018), ARC-challenge (Clark et al., 2018), SciQ (Welbl et al., 2017), HellaSwag (Zellers et al., 2019), RACE (Lai et al., 2017) | GitHub |
| **BoLT** (Ruan et al., 2025) | Reasoning to learn , synthetic latent thoughts, Expectation-Maximization (EM) algorithm, Monte Carlo Sampling | TinyLLaMA (Zhang et al., 2024), GPT-4o-mini (Hurst et al., 2024) | Mathematical reasoning, scientific and logical reasoning | MATH (Hendrycks et al., 2021b), GSM8K (Cobbe et al., 2021), MMLU-STEM (Hendrycks et al., 2021a); FineMath-4+ (Lozhkov et al.) | GitHub |
| **LaTRO** (Chen et al., 2024a) | Formulates reasoning as sampling from a latent distribution and optimizes it via variational approaches | Phi-3.5-mini (Abdin et al., 2024), Mistral-7B (Jiang et al., 2023), LLaMA-3.1-8B (Grattafiori et al., 2024) | Mathematic reasoning, logic reasoning | GSM8K (Cobbe et al., 2021), ARC-challenge (Clark et al., 2018) | GitHub |
| **Soft Thinking** (Zhang et al., 2025c) | Training-free, emulates human-like 'soft' reasoning by generating soft, abstract concept tokens in a continuous concept space, concept token, cold stop | QwQ-32B (Team, 2025), DeepSeek-R1-Distill-Qwen-32B (Guo et al., 2025), DeepSeek-R1-Distill-LLaMA-70B (Guo et al., 2025) | Mathematical reasoning, programming (coding) | MATH-500 (Lightman et al., 2024), AIME24 (MathAI, 2024a), GSM8K (Cobbe et al., 2021), GPQA-Diamond (Rein et al., 2024), HumanEval (Chen et al., 2021), MBPP (Austin et al., 2021), LiveCodeBench (Jain et al., 2025) | GitHub |
| **SoftCoT** (Xu et al., 2025b), | Soft thought tokens, a lightweight fixed assistant model, continuous-space reasoning, soft prompt tuning | Qwen2.5-7B-Instruct (Yang et al., 2024b), LLaMA-3.1-8B-Instruct (Grattafiori et al., 2024), Qwen3-8B (Yang et al., 2025a) | Mathematical reasoning, commonsense reasoning, symbolic reasoning | GSM8K (Cobbe et al., 2021), AQUA-RAT (Ling et al., 2017), StrategyQA (Geva et al., 2021), Date Understanding (Srivastava et al., 2023), ASDiv-Aug (Xu et al., 2025b) | GitHub |
| **SoftCoT++** (Xu et al., 2025a) | Test-time scaling, continuous latent space, contrastive learning, | LLaMA-3.1-8B-Instruct (Grattafiori et al., 2024), Qwen3-8B (Yang et al., 2025a) | Mathematical reasoning, commonsense reasoning, symbolic reasoning | GSM8K (Cobbe et al., 2021), ASDiv-Aug (Xu et al., 2025b), AQUA-RAT (Ling et al., 2017), StrategyQA (Geva et al., 2021), Date Understanding (Srivastava et al., 2023) | GitHub |
| **COT2** (Gozeten et al., 2025) | Explicitly track multiple traces in parallel, GRPO-based policy optimization, MTS (multi-token sampling) | GPT-2 (Radford et al., 2019) | MNNS (Minimum Non-Negative Sum), logical reasoning, multi-hop commonsense reasoning | ProntoQA (Saparov & He, 2023), ProsQA (Hao et al., 2024) | **-** |

ing (Cheng & Van Durme, 2024; Shen et al., 2025b), providing weak alignment and leading to suboptimal performance. To address this, *SynAdapt* (Wang et al., 2025a) introduces synthetic continuous chain-of-thought representations as full alignment targets, enabling iterative refinement of draft trajectories without autoregressive generation. It further integrates a difficulty classifier to adaptively route easy questions to efficient latent reasoning while prompting explicit CoT re-thinking on harder ones, achieving a better balance between accuracy and efficiency.

**(b) Adaptive Efficiency.** This group of methods targets dynamic or adaptive trajectory compression during reasoning to dynamically adjust reasoning length (Ma et al., 2025) or speed (Tan et al., 2025), reducing redundant reasoning and enabling adaptive reasoning efficiency while maintaining accuracy (Zhang et al., 2025a). Particularly, *LightThinker* (Zhang et al., 2025a) dynamically compresses intermediate reasoning steps into compact gist tokens after a fixed number of tokens or a complete semantic segment, discarding verbose reasoning traces in favor of compact representations, and thereby reducing context length while preserving reasoning continuity and task performance. *CoT-Valve* (Ma et al., 2025) enables elastic control over reasoning length by identifying a direction in parameter space. It allows a single model to dynamically generate variable-length reasoning traces based on task difficulty, and further supports progressive reasoning compression. *Compressed Latent Reasoning (CoLaR)* (Tan et al., 2025) compresses reasoning chains into latent space via auxiliary next compressed embedding prediction and enhances the diversity of latent trajectories through a non-deterministic latent head and GRPO (Shao et al., 2024; Yu et al., 2025b)-based reinforcement learning. Importantly, CoLaR allows dynamic control over reasoning length and speed at inference time by easily prompting the compression factor.

**(c) Progressive Refinement.** This line of work refines the implicit reasoning process progressively by step-by-step internalization or iterative updating. The former internalizes explicit reasoning steps into latent reasoning step by step, ensuring a smooth transition from explicit reasoning to implicit reasoning (Deng et al., 2024; Hao et al., 2024; Shen et al., 2025a). The latter progressively refines latent representations by multiple iterative steps during pretraining, improving reasoning performance (Zeng et al., 2025; Ruan et al., 2025).

Inspired by curriculum learning, *ICoT-SI* (Deng et al., 2024) proposes an innovative stepwise internalization strategy by gradually removing the explicit CoT tokens and fine-tuning to predict the remaining tokens until the model can generate answers directly from the input. *Chain-of-Continuous-Thought (Coconut)* (Hao et al., 2024) treats the last hidden states as continuous thoughts, and progressively replaces the CoT steps with these thoughts through curriculum training, exploring a fully differentiable latent reasoning paradigm and supporting breadth-first search over multiple latent steps. Similar to *Coconut*, *Heima* (Shen et al., 2025a) gradually replaces entire reasoning chains with "thinking tokens" via a dedicated encoder. For interpretability, *Heima* also employs an adaptive decoding based on standard LLMs to reconstruct variable-length CoTs from the last hidden representations of thinking tokens. *PonderingLM* (Zeng et al., 2025) integrates a pondering mechanism into language models by iteratively feeding back a weighted sum of all token embeddings into the input across multiple forward passes within a single generation step. And it enables fully differentiable and self-supervised refinement without discrete sampling or human annotations, providing a new scaling via pondering steps. *BoLT* (Ruan et al., 2025) explicitly infers latent thoughts underlying the data generation process and performs reasoning from these thoughts, improving pretraining data efficiency and enabling self-bootstrapping performance via the EM-style iterations.

**(d) Exploratory Diversification.** Explicit reasoning usually samples from a finite vocabulary, restricting exploration to a single reasoning trajectory and offering limited information capacity (Zhang et al., 2025c; Xu et al., 2025b; Gozeten et al., 2025). Instead, exploratory-style implicit reasoning methods introduce soft or perturbed latent representations into the model's latent space through sampling from latent space or probabilistic mixtures (Chen et al., 2024a; Gozeten et al., 2025; Zhang et al., 2025c). These methods broaden the exploratory space and promote the diversity of possible reasoning trajectories while preserving compatibility with LLM backbones (Xu et al., 2025b).

In particular, *Latent Reasoning Optimization (LaTRO)* (Chen et al., 2024a) formulates the reasoning process as sampling latent trajectories and optimizes their distribution via self-rewarding under a variational framework. By maximizing their likelihood of correct answers given the sampled trajectories, LaTRO improves

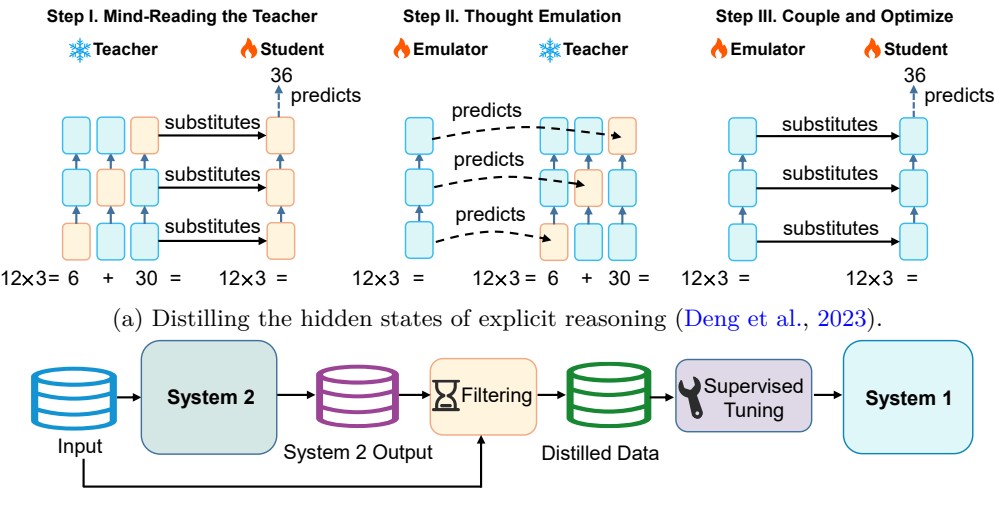

(a) Distilling the hidden states of explicit reasoning (Deng et al., 2023).

(b) Distilling the data from System 1 into System 2 (Yu et al., 2024b).

Figure 5: Two representative distillation methods of internal-state-level latent optimization.

the quality of reasoning trajectories without external feedback. *Soft Thinking* (Zhang et al., 2025c) generates probabilistically weighted concept tokens that represent mixtures of discrete semantics, allowing the model to implicitly explore multiple reasoning trajectories in parallel and enabling training-free reasoning in a continuous concept space. Furthermore, *SoftCoT* (Xu et al., 2025b) injects instance-specific continuous latent tokens generated by a lightweight assistant model into the target LLM's embedding space, enabling soft chain-of-thought reasoning without modifying the backbone or inducing catastrophic forgetting and enriching the probability space for exploration. *SoftCoT++* (Xu et al., 2025a) extends soft chain-of-thought reasoning to the test-time scaling paradigm. It perturbs latent thoughts with multiple specialized initial tokens and employs a contrastive learning objective to generate diverse reasoning trajectories in continuous latent space, achieving robust performance across diverse reasoning tasks. *COT2* (Gozeten et al., 2025) also allows models to explore multiple reasoning trajectories in parallel by continuously-valued tokens. It introduces a continuous supervision strategy that aligns softmax outputs with empirical token distributions, and proposes multi-token sampling and GRPO-based (Shao et al., 2024; Yu et al., 2025b) policy optimization.

### 3.1.3 Internal-State-Level

Internal-state-level latent optimization methods take the model's **internal states** as the target of reasoning regulation. They transform explicit CoT supervision into latent embeddings (Deng et al., 2023), distill structured reasoning into compact internal representations (Wang et al., 2025c; Yu et al., 2024b), or support implicit computation through memory and posterior inference modules (Orlicki, 2025; Kong et al., 2025). Some methods further integrate internal-state optimization into downstream tasks such as recommendation (Tang et al., 2025). Figure 5 illustrates representative approaches, and Table 4 summarizes their key information.

Deng et al. (2023) propose *ICoT-KD*, which enables implicit reasoning by distilling hidden states from a horizontal CoT teacher into a student. An emulator is introduced to predict the teacher's intermediate hidden states, and coupled with the student for end-to-end training, allowing the student to perform vertical reasoning directly in the hidden state space without explicit CoT steps. Yu et al. (2024b) distill System 2 with intermediate outputs into System 1 without intermediate outputs. They filter the training set of System 2 based on the self-consistency of outputs and self-consistency under input perturbation to fine-tune the LLM into System 1 with supervision. *ReaRec* (Tang et al., 2025) autoregressively feeds the last hidden state back into the model for implicit multi-step reasoning, pioneering the integration of inference-time computation into sequential recommendation. Specifically, *ReaRec* proposes two strategies: Ensemble Reasoning Learning (ERL), which draws on ensemble learning to capture latent interest distributions, and Progressive Reasoning Learning (PRL), which incorporates curriculum learning via progressive temperature annealing to gradually

Table 4: Internal-state-level latent optimization (§3.1.3).

| Models | Main Techniques | Base Model | Main Tasks, Scenarios | Main Datasets | Open Source |
|---|---|---|---|---|---|
| **ICoT-KD** (Deng et al., 2023) | Knowledge distillation, a three-step approach: mind-reading the teacher, thought emulation, coupling and optimization | GPT-2 Small (Radford et al., 2019), GPT-2 Medium (Radford et al., 2019) | Multi-digit multiplication, Grade school math problem | BIG-Bench (Srivastava et al., 2023), GSM8K-Aug (Deng et al., 2023) | GitHub |
| **System2 Distillation** (Yu et al., 2024b) | Distillation data, supervised fine-tuning, unsupervised consistency criterion | LLaMA-2-70B-Chat (Touvron et al., 2023b) | Symbolic reasoning, Syco-phancyEval QA, LLM-as-judge, Math reasoning | Last letter concatenation, Coin flip, TriviaQA (Joshi et al., 2017), OASST2 (Köpf et al., 2023), MT-bench (Zheng et al., 2023), GSM8K (Cobbe et al., 2021) | - |
| **ReaRec** (Tang et al., 2025) | Inference-time computation, multi-step implicit reasoning, ensemble reasoning learning (ERL), progressive reasoning learning (PRL), think-before-action | Model-agnostic Transformer | Sequential recommendation | Yelp (Yelp, 2024), Amazon 2023 (Hou et al., 2024) | GitHub |
| **Beyond Words** (Orlicki, 2025) | Implicit mental representation, memory approach, implicit memory module (IMM), memory write, memory read | nanoGPT (Karpathy, 2023) | Language modeling | Shakespeare (Karpathy, 2023) | - |
| **System-1.5 Reasoning** (Wang et al., 2025c) | Depth shortcut (DS) vertically, step shortcut (SS) horizontally, budget-controllable test-time scaling, knowledge distillation | GPT-2 Small (Radford et al., 2019), LLaMA-3.2-1B (Meta, 2024) | Mathematical reasoning, common sense reasoning | GSM8K-Aug (Deng et al., 2023), GSM-Hard (Gao et al., 2023), StrategyQA (Geva et al., 2021) | - |
| **LTMs** (Kong et al., 2025) | Latent thought vectors, variational Bayes, dual-rate optimization, fast-slow learning | Training from scratch | Zero-shot perplexity evaluation, arithmetic reasoning, conditional generation, unconditional generation | Penn Treebank (PTB) (Marcus et al., 1993), WikiText (Merity et al., 2017), One Billion Word Benchmark (Chelba et al., 2013), LAM-BADA (Paperno et al., 2016), AG News (Zhang et al., 2015), PubMed (Cohan et al., 2018), arXiv (Cohan et al., 2018), GSM8K (Cobbe et al., 2021); OpenWebText (Gokaslan & Cohen, 2019) | - |
| **HRPO** (Yue et al., 2025b) | Reinforcement learning, hybrid latent reasoning, a learnable gating mechanism, stochastic token sampling | Qwen2.5-1.5B-Instruct (Yang et al., 2024b), Qwen2.5-3B-Instruct (Yang et al., 2024b) | Open-domain and multi-hop question answering, STEM benchmarks | Natural Questions (Kwiatkowski et al., 2019), TriviaQA (Joshi et al., 2017), HotpotQA (Yang et al., 2018), 2WikiMultiHopQA (Ho et al., 2020), Bamboogle (Press et al., 2023), GSM8K (Cobbe et al., 2021), MATH (Hendrycks et al., 2021b), MATH-500 (Lightman et al., 2024), MMLU-STEM (Hendrycks et al., 2021a), ARC-challenge (Clark et al., 2018) | GitHub |

refine hidden state distributions. *Beyond Words* (Orlicki, 2025) views the summaries of hidden states as implicit mental representations which are dynamically stored and retrieved by an Implicit Memory Module (IMM), capturing reasoning-related past context and sensory-like memory for internal reasoning. *System-1.5 Reasoning* (Wang et al., 2025c) proposes dynamic shortcuts and introduces router–adapter modules at each Transformer layer after language-to-latent distillation. By the trained router, vertical depth shortcuts enable non-critical steps to exit early and critical steps to deeper layers, and then horizontal step shortcuts directly copy hidden states at early-exit points to skip trivial steps. *Latent Thought Models (LTMs)* (Kong et al., 2025) incorporate latent thought vectors sampled from a Gaussian prior into Transformer layers by cross-attention. These latent vectors are optimized by fast and slow learning and serve as abstracts of the entire sequence, guiding autoregressive generation and enabling new scaling behaviors (i.e., inference steps and latent thought size). To enable a hybrid latent reasoning on discrete and continuous representations, HRPO (Yue et al., 2025b) introduces a gating mechanism that progressively incorporates hidden states into sampled token embeddings, producing hybrid rollouts. To optimize such rollouts, it leverages reinforcement learning with outcome-based rewards, enabling latent reasoning without CoT supervision.

## 3.2 Signal-Guided Control

Signal-guided control methods steer internal reasoning by inserting specialized tokens that modulate computation without producing intermediate textual outputs. This strategy offers a lightweight and architecture-compatible mechanism of enabling controllable and interpretable reasoning. Based on the design and functionality of the inserted control signals, existing approaches can be broadly categorized into single-type signal methods (§3.2.1) and multi-type signal methods (§3.2.2). See Table 5 for a comprehensive overview.

### 3.2.1 Single-Type Signal

Single-type signal denotes a single control mechanism that uniformly modulates the reasoning process, realized either by inserting an explicit control token (e.g., thinking tokens (Herel & Mikolov, 2024), planning tokens (Wang et al., 2024c)) or by token-free latent control that adjusts latent embeddings or intermediate states (e.g., LatentSeek (Li et al., 2025a)). These signals (Herel & Mikolov, 2024; Goyal et al., 2024; Pfau et al., 2024; Li et al., 2025a; Kim et al., 2025) act as lightweight computational markers to adjust reasoning dynamics, often without requiring architecture changes or external supervision. By introducing these signals either statically during training or adaptively at test time, models can allocate additional internal computation to uncertain or complex inputs, improving reasoning flexibility and generalization.

One class of approaches statically injects predefined or learnable tokens into the input sequence to allocate additional reasoning capacity, thereby extending inference time and depth in a uniform manner. Representative examples include the use of *thinking* token (Herel & Mikolov, 2024), *pause* token (Goyal et al., 2024), thought token (Zelikman et al., 2024), *filler* token (Pfau et al., 2024), and *planning* token (Wang et al., 2024c). Particularly, Herel & Mikolov (2024) add *thinking tokens* after each word, providing more time and computations for complex tasks and improving the generalization capability of RNN-based language models without architectural modification or supervision. In parallel, Goyal et al. (2024) explore the new paradigm, named delayed next-token prediction, and append learnable *pause tokens* to input sequences during training and inference, delaying the answer outputs until the last pause token is processed. This design introduces wider computational pathways by inserting *pause tokens*, enabling the model to internally 'think longer'. *Quiet-STaR* (Zelikman et al., 2024) learns to reason generally from text data. It generates internal rationales at every token in parallel by attention mask and introduces learned meta-tokens to control the rationale generation. Furthermore, it uses a non-myopic loss and mixing residual head for effective reasoning and mitigating early distribution shift, respectively. To better control reasoning generation in LLMs, Pfau et al. (2024) shows that transformers can use meaningless *filler tokens* to replace CoT tokens, like '......' or '.'. It also highlights that *filler tokens* require specific, dense supervision and can improve performance in parallelizable tasks. Additionally, Wang et al. (2024c) introduce *planning tokens* as the high-level plan of current reasoning step to guide useful reasoning steps generation. The LLM generates *planning tokens* before reasoning steps by three alternative ways (i.e., Arithmetic, K-Means, and SQ-VAE), improving model performance especially for long reasoning scenarios due to augmented computational space and learned specialization by *planning tokens*. In contrast to fixed token insertion, recent methods such as LatentSeek (Li et al., 2025a) and DIT (Kim et al., 2025) dynamically adjust embeddings or token placement during inference, enabling instance-aware latent control and enhancing reasoning. *LatentSeek* (Li et al., 2025a) introduces a novel test-time instance-level adaptation framework that iteratively optimizes token-wise latent representations via self-rewarding policy gradient at test time. The latent representations control and guide better reasoning paths for each problem instance without parameter updates. Similarly, *Dynamic Inserting Tokens Training (DIT)* (Kim et al., 2025) proposes a log-likelihood-based method to insert $[PAUSE]$ tokens at positions of low model confidence, identified via token-level log-probability. These dummy tokens trigger additional internal computation without emitting output, enhancing the model's ability to predict subsequent low-probability tokens.

### 3.2.2 Multi-Type Signal

Multi-type signal methods employ multiple distinct control signals (Jin et al., 2024; Fang et al., 2025a), each governing a specific aspect of the reasoning process. Compared with single-type mechanisms, these methods enable finer-grained control over reasoning behaviors, offering more structured organization and adaptive adjustment to different reasoning demands.

*Memory & Reasoning* (Jin et al., 2024) proposes a novel LLM inference paradigm that decomposes the inference process into two explicit actions: memory recall and reasoning, guided by learnable control tokens $<memory>$ and $<reason>$, thereby improving both performance and interpretability through structured response generation. Similarly, *Thinkless* (Fang et al., 2025a) enables LLMs to adaptively choose between short-form and long-form inference via two control tokens $<short>$ and $<think>$, and introduces Decoupled GRPO (Shao et al., 2024) (DeGRPO) to optimize mode selection and answer generation separately.

Table 5: Signal-guided control (§3.2).

| Models | Main Techniques | Base Model | Main Tasks, Scenarios | Main Datasets | Open Source |
|---|---|---|---|---|---|
| **thinking tokens** (Herel & Mikolov, 2024) | Running more time and calculations for complex problems, unsupervised learning | LSTM LM (Graves & Graves, 2012) | Language modeling, mathematical reasoning | Penn Treebank (PTB) (Marcus et al., 1993), WikiText-2 (Merity et al., 2017), Mathematics Dataset (Saxton et al., 2019), MacroEconomics textbook (Cooper & John, 2013) | - |
| **pause tokens** (Goyal et al., 2024) | Delayed next-token prediction, pause-pretraining, pause-finetuning, pause-inference | Decoder-only Transformer | Reasoning task, extractive question answering, general understanding, long term context recall, natural language inference, fact recall | GSM8K (Cobbe et al., 2021), SQuAD (Rajpurkar et al., 2016), CoQA (Reddy et al., 2019), CommonsenseQA (Talmor et al., 2019), PIQA (Bisk et al., 2020), LAMBADA (Paperno et al., 2016), HellaSwag (Zellers et al., 2019), WebQuestions (Berant et al., 2013), Natural Questions (Kwiatkowski et al., 2019); C4 (Raffel et al., 2020) | - |
| **Quiet-STaR** (Zelikman et al., 2024) | Generate rationales in parallel, thought token, mixing residual heads, a teacher-forcing trick, reinforcement learning | Mistral-7B (Jiang et al., 2023) | Zero-shot reasoning | CommonsenseQA (Talmor et al., 2019), GSM8K (Cobbe et al., 2021); OpenWebMath (Paster et al., 2024), C4 (Raffel et al., 2020) | GitHub |
| **filler tokens** (Pfau et al., 2024) | Providing hidden computations | LLaMA (Touvron et al., 2023a) 34M | 3SUM, 2SUM | Synthetic data: 3SUM, 2SUM | GitHub |
| **planning tokens** (Wang et al., 2024c) | Generic prefix planning tokens, special planning tokens, Arithmetic, K-Means, SQ-VAE | Phi-1.5 (Li et al., 2023b), LLaMA-2-7B (Touvron et al., 2023b), LLaMA-2-13B (Touvron et al., 2023b) | Math word problem, multihop QA | GSM8K (Cobbe et al., 2021), AQUA-RAT (Ling et al., 2017), MATH (Hendrycks et al., 2021b), StrategyQA (Geva et al., 2021) | GitHub |
| **LatentSeek** (Li et al., 2025a) | Test-Time Instance-level Adaptation (TTIA), iteratively refining latent representations, continuous latent space, reinforcement learning | Qwen2-7B-Instruct (Yang et al., 2024a), Qwen2.5-1.5B-Instruct (Yang et al., 2024b), Qwen2.5-7B-Instruct (Yang et al., 2024b), Qwen2.5-14B-Instruct (Yang et al., 2024b), LLaMA-3.1-8B-Instruct (Grattafiori et al., 2024), Mistral-7B-Instruct-v0.3 (Jiang et al., 2023) | Mathematical reasoning | GSM8K (Cobbe et al., 2021), MATH-500 (Lightman et al., 2024), AIME24 (MathAI, 2024a) | GitHub |
| **DIT** (Kim et al., 2025) | Identifies positions within sequences where model confidence is lowest, log-likelihood-based [PAUSE] tokens inserting | Phi-2-2.7B (Javaheripi et al., 2023), Phi-3-mini (Abdin et al., 2024), LLaMA-3-8B (Grattafiori et al., 2024) | Mathematical reasoning, code reasoning | GSM8K (Cobbe et al., 2021), AQUA-RAT (Ling et al., 2017), MBPP (Austin et al., 2021) | GitHub |
| **Memory & Reasoning** (Jin et al., 2024) | Disentangles memory and reasoning ability, two special tokens $<memory>$ and $<reason>$ | LLaMA-2-7B-Chat (Touvron et al., 2023b), LLaMA-3.1-8B-Instruct (Grattafiori et al., 2024), Qwen2.5-7B-Instruct (Yang et al., 2024b), GPT-4o, GPT-4o-mini (Hurst et al., 2024) | Multi-hop QA, commonsense reasoning, fact verification | StrategyQA (Geva et al., 2021), CommonsenseQA (Talmor et al., 2019), TruthfulQA (Lin et al., 2022) | GitHub |
| **Thinkless** (Fang et al., 2025a) | LLM learns when to think, adaptively select between short-form and long-form reasoning, Decoupled GRPO (DeGRPO), RL, control token ($<short>$, $<think>$) and response token | DeepSeek-R1-Distill-Qwen-1.5B (Guo et al., 2025) | Mathematical reasoning | AIME24 (MathAI, 2024a), Minerva Algebra (MATH (Hendrycks et al., 2021b)), MATH-500 (Lightman et al., 2024), GSM8K (Cobbe et al., 2021) | GitHub |

### 3.3 Layer-Recurrent Execution

Layer-recurrent execution introduces recurrence into the forward computation of transformer models, enabling multi-step reasoning through repeated internal computation, as shown in Figure 6. Similar to expanding model depth, these methods reuse weights across layers (or blocks) to iteratively refine token representations (Chen et al., 2025c; Saunshi et al., 2025; Mohtashami et al., 2025; Geiping et al., 2025; Yu et al., 2025a). This enables fine-grained and token-adaptive computation while preserving parameter efficiency, allowing LLMs to simulate deep reasoning chains internally and achieve generalization in long-context or multi-hop tasks. See Table 6 for a comprehensive overview about the key information of these methods.

To realize such recurrent computation in practice, several studies develop transformer variants that simulate multi-step reasoning by iteratively refining token representations through shared weights and dynamic depth control (Chen et al., 2025c; Saunshi et al., 2025; Mohtashami et al., 2025). More precisely, *Inner Thinking Transformer (ITT)* (Chen et al., 2025c) formulates token generation of reasoning as multiple implicit thinking steps in a dynamic token-wise depth architecture without parameter increase. By adaptive token routing networks, *ITT* selects critical tokens of inner thinking layers to allocate additional thinking steps for deeper thinking. It also iteratively refines tokens' representations by accumulating the residual of each inner thinking step. In parallel, Saunshi et al. (2025) show that a *looped Transformer*, which achieves large depth through looping while maintaining parameter efficiency via weight sharing, can effectively solve

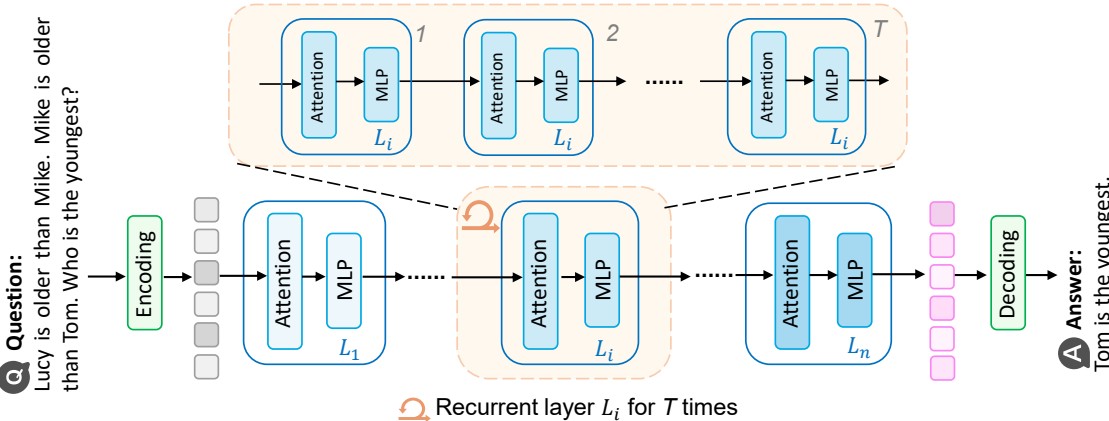

Figure 6: Simplified illustration of layer-recurrent execution for implicit reasoning, which usually reuses the same parameters at recurrent layers (or blocks) to refine reasoning through depth-wise computation.

reasoning tasks. They further demonstrate that such models can simulate $T$-step CoT reasoning through $T$ loops by implicitly generating latent thoughts in parallel. To enhance reasoning without degrading perplexity, they also introduce a looping-based regularization that encourages similarity across layer weights using cosine similarity. Similarly, *CoTFormer* (Mohtashami et al., 2025) builds on the distinction between CoT and iteratively applying the model multiple times, and recurrently uses a deeper Transformer with weight tying. For computation-accuracy trade-off like *ITT*, *CoTFormer* dynamically varies the number of re-uses by token-wise adaptive repeats for the different difficulties of tokens. Another direction focuses on improving the fidelity and scalability of loop-based reasoning by aligning recurrent computation with explicit reasoning steps or by expanding test-time compute capacity (Geiping et al., 2025; Yu et al., 2025a). Specifically, *Huginn* (Geiping et al., 2025) designs a depth-recurrent model consisting of a prelude block for encoding, a core shared recurrent block for iterative latent-state computation, and a coda block for decoding. *Huginn* feeds inputs repeatedly into each step and randomly initializes latent states for path independence. During training, *Huginn* randomly samples iteration counts from a log-normal Poisson distribution for scaling up test-time iterations, and adopts truncated backpropagation for efficient optimization where gradient updates are limited to the last $k$ iterations. To mitigate the low accuracy issue of explicit reasoning for long sequence reasoning, *RELAY* (Yu et al., 2025a) aligns the iteration of looped models with the stepwise reasoning of CoT by the proposed iteration-wise alignment mechanism. The trained looped model with length generalization can generate accurate reasoning chains for complex problems, which are regarded as high-quality data to fine-tune an auto-regressive model.

## 4 Mechanistic and Behavioral Evidence

Although numerous recent studies have introduced approaches to leverage or enhance implicit reasoning in LLMs, the existence and nature of such latent reasoning processes remain subjects of ongoing investigation. This section presents a structured review of empirical and mechanistic evidence indicative of implicit reasoning in LLMs. The discussion is organized into three complementary perspectives: structural patterns identified in intermediate model layers (§4.1), behavioral signatures manifested during inference (§4.2), and representation-level findings derived from probing and intervention methodologies (§4.3).

### 4.1 Layer-wise Structural Evidence

This line of evidence investigates whether LLMs perform implicit reasoning by analyzing structural patterns that emerge across model layers. Several studies demonstrate that the activations of intermediate layers can approximate final outputs (Din et al., 2024), or encode task-specific subtasks at different depths of layers (Yang et al., 2025b). Others provide theoretical constructions illustrating how transformer layers can support implicit iterative computation by directed graphs (Zhu et al., 2025a; Xu & Sato, 2025). Col-

Table 6: Layer-recurrent execution (§3.3).

| Models | Main Techniques | Base Model | Main Tasks, Scenarios | Main Datasets | Open Source |
|---|---|---|---|---|---|
| **ITT** (Chen et al., 2025c) | Adaptive Token Routing, Thinking Step Encoding, Residual Thinking Connection | LLaMA-2 (Touvron et al., 2023b) architecture | Common sense and reading comprehension, continued QA and text understanding | SciQ (Welbl et al., 2017), PIQA (Bisk et al., 2020), Wino-Grande (Sakaguchi et al., 2020), ARC-easy (Clark et al., 2018), HellaSwag (Zellers et al., 2019), ARC-challenge (Clark et al., 2018), LogiQA (Liu et al., 2021), BoolQ (Clark et al., 2019), LAMBADA (Paperno et al., 2016); RedPajama (Weber et al., 2024) | - |
| **looped Transformer** (Saunshi et al., 2025) | K-layer transformer looped L times, looping-based regularization, simulating CoT reasoning | Decoder-only Transformer | N-ary addition, p-hop induction, synthetic grade school math problems, closed book QA, open book QA, math word problems, reasoning primitives | TriviaQA (Joshi et al., 2017), TydiQA-NoContext (Clark et al., 2020), Natural Questions (Kwiatkowski et al., 2019), ComplexWebQuestions (Talmor & Berant, 2018), TydiQA-GoldP (Clark et al., 2020), SQuAD 2.0 (Rajpurkar et al., 2018), DROP (Dua et al., 2019), QuAC (Choi et al., 2018), CoQA (Reddy et al., 2019), SVAMP (Patel et al., 2021), AS-Div (Miao et al., 2020), MAWPS (Koncel-Kedziorski et al., 2016); Pile (Gao et al., 2020) | - |
| **CoTFormer** (Mohtashami et al., 2025) | A compute adaptive model, token-wise adaptive repeats | Pre-LayerNorm Transformer architecture (Xiong et al., 2020) | Zero-shot reasoning | MMLU (Hendrycks et al., 2021a), ARC (Clark et al., 2018), HellaSwag (Zellers et al., 2019), PIQA Bisk et al. (2020); OpenWebText2 (Gao et al., 2020) | GitHub |
| **Huginn** (Geiping et al., 2025) | A latent recurrent-depth architecture, test-time scaling, truncated backpropagation | Decoder-only Transformer | Lm-eval-harness tasks, mathematical reasoning and understanding, code reasoning | ARC-easy (Clark et al., 2018), ARC-challenge (Clark et al., 2018), HellaSwag (Zellers et al., 2019), MMLU (Hendrycks et al., 2021a), OpenBookQA (Mihaylov et al., 2018), PIQA (Bisk et al., 2020), SciQ (Welbl et al., 2017), Wino-Grande (Sakaguchi et al., 2020), GSM8K (Cobbe et al., 2021), MATH (Hendrycks et al., 2021b), MathQA (Amini et al., 2019), MBPP (Austin et al., 2021), HumanEval (Chen et al., 2021) | GitHub |
| **RELAY** (Yu et al., 2025a) | Looped Transformer with length generalization, iteration-wise alignment with CoT, multitask learning | Encoder-only Transformer | Arithmetic, Edit Distance (ED), Longest Increasing Subsequence (LIS) | Self-constructed datasets | GitHub |

lectively, these studies offer mechanistic insights into how reasoning may be realized through depth-wise transformations, latent trajectory formation, and structural reuse within standard architectures.

Concretely, *Jump to Conclusions* (Din et al., 2024) reveals that linear projections from intermediate layers can approximate final predictions with high precision. This provides structural evidence that reasoning may be completed internally without requiring full-depth processing. Lin et al. (2025) find that language models trained on fixed-pattern mathematical reasoning data can achieve high accuracy via implicit reasoning, yet fail to generalize when trained on unfixed-pattern data. They also trace the information flow across layers, and argue that implicit reasoning arises primarily through shortcut learning rather than robust generalization, particularly for the unfixed pattern. *Internal Chain-of-Thought* (Yang et al., 2025b) claims that LLMs sequentially decompose and execute composite tasks across layers, where distinct subtasks are learned at different depths and performed in order. This study also reveals a consistent layer-wise execution pattern by LogitLens decoding, providing mechanistic evidence for internal planning in LLMs. *Reasoning by Superposition* (Zhu et al., 2025a) presents a theoretical construction showing that a two-layer transformer can solve graph reachability problems through $D$ steps of continuous thoughts, where superposition states encode multiple implicit search traces simultaneously. This construction aligns closely with the solutions discovered via training dynamics. *To CoT or To Loop* (Xu & Sato, 2025) provides structural evidence for LLM implicit reasoning by analyzing the computation process of looped Transformers through directed acyclic graphs (DAGs). It is shown that looped Transformers can simulate DAGs layer by layer, enabling efficient parallel reasoning on deterministic tasks in contrast to the explicit token-level inference of CoT.

## 4.2 Behavioral Signatures

Another line of investigation focuses on observable behaviors exhibited by LLMs to infer the presence of latent reasoning processes. By analyzing training dynamics, response patterns, and other behavioral signatures, these studies aim to determine whether LLMs internally compute reasoning steps without explicitly emitting them. For example, Wang et al. (2024a) show that extended training can induce a phase transition from memorization to generalization, enabling implicit reasoning to emerge. Additional evidence stems from exploring step skipping (Liu et al., 2024b) and reasoning leaps behaviors (Hagendorff & Fabi, 2025), which reveal the model's capacity to internalize computations and flexibly adjust reasoning granularity.

Specifically, Wang et al. (2024a) introduce a *Grokked Transformer* and reveal that the transformer can robustly acquire implicit reasoning abilities through extended training far beyond overfitting, known as the *grokking phenomenon*, during which the model transitions from memorizing circuits to generalizing circuits. Their findings also uncover that the data distribution (i.e., the ratio between inferred and atomic facts), not data size, is the key to generalization. Yang et al. (2024c) explore latent multi-hop reasoning of LLMs using one-hop and two-hop prompts, evaluating whether the models internally recall the bridge entity and measuring the consistency of response outputs between one-hop and two-hop prompts. Liu et al. (2024b) investigate the *step-skipping* behavior of LMs, enabling reasoning in fewer steps by fine-tuning LLMs via mixed datasets that include full-step reasoning paths and self-generated step-skipping paths. This implies that some steps can be internalized and skipped during reasoning without sacrificing accuracy. Hagendorff & Fabi (2025) quantify the capacities of reasoning leaps between individual tokens by designing non-English language responses to benchmark implicit reasoning of 18 LLMs, demonstrating that the models engage in genuine internal reasoning rather than relying solely on heuristics, especially for dense models.

### 4.3 Representation-Based Analysis

The third line of evidence focuses on internal representations, aiming to determine whether LLMs encode reasoning processes in their hidden states or activation dynamics. By leveraging probing methods, activation interventions or mechanistic reverse-engineering, these studies examine how latent reasoning manifests in geometric and functional properties of the representation space. For example, Hou et al. (2023) reveal that reasoning trees can be detected from the model's attentions, while CoE (Wang et al., 2024d) analyzes directional changes in hidden trajectories to evaluate inference quality. Further evidence comes from activation space perturbation to elicit reasoning (Zhang & Viteri, 2025) and dissecting symbolic inference circuits (Brinkmann et al., 2024), offering deeper insight into the mechanisms underlying implicit reasoning.

In particular, *MechanisticProbe* (Hou et al., 2023) reveals that language models implicitly encode reasoning trees within their attention patterns by designing a new probing approach, providing mechanistic evidence that LMs indeed internally perform multi-step reasoning. *TTT* (Kudo et al., 2024) investigates the internal reasoning of LMs by causal probing and intervention, finding that single subproblems are resolved in a post-hoc Think-to-Talk mode where the reasoning is finished and answers are determined before CoT begins, while complex multi-step problems are resolved in a step-by-step Talk-to-Think mode during CoT. Yu (2024) also investigates whether implicit reasoning really calculates the intermediate results by linearly probing hidden states, finding that trained implicit CoT indeed calculates these results, but prompted implicit CoT hardly does. *Distributional Reasoning* (Shalev et al., 2024) reveals that LLMs implicitly perform multi-hop inference by distributing multiple potential intermediate answers in the activation of intermediate states, implying parallel reasoning paths in implicit multi-hop reasoning. *CoE* (Wang et al., 2024d) regards progressive hidden states as latent thinking paths and studies the dynamic magnitude and angle changes of paths to evaluate the correctness of reasoning responses, indirectly supporting that reasoning information exists in hidden states. Brinkmann et al. (2024) study the internal mechanisms of reasoning by reverse-engineering a transformer trained on a symbolic multi-step reasoning task, revealing that the model implements a depth-bounded recurrent mechanism within its internal representations, and performs symbolic reasoning by *backward chaining* algorithm without the aid of CoT. Zhang & Viteri (2025) design a *steering vector intervention* approach in the activation space to induce reasoning without relying on explicit natural language prompting, suggesting that reasoning patterns can be implicitly encoded into network weights and activations.

## 5 Evaluation and Benchmark

Despite increasing interest in LLM implicit reasoning, the evaluation of such methods remains underdeveloped. Unlike explicit reasoning which exposes intermediate steps for inspection and error localization, implicit reasoning operates entirely within the model's internal states, posing new challenges for measurement, interpretability, and comparison. This section outlines existing evaluation practices, including commonly used metrics (§5.1) and benchmark datasets (§5.2), and presents their roles in capturing the full reasoning capabilities of implicit methods.

## 5.1 Metrics

In this section, we review commonly used metrics for evaluating implicit reasoning methods, and categorize them into four key dimensions, covering output correctness, resource efficiency, underlying language modeling capabilities and internal probing. These dimensions collectively provide complementary perspectives, enabling a more comprehensive assessment of answer correctness (§5.1.1), resource efficiency (§5.1.2), perplexity (§5.1.3), and probing accuracy (§5.1.4).

### 5.1.1 Answer Correctness

Implicit reasoning evaluation typically focuses on end-task answers, using final answer correctness and quality as a proxy for reasoning success. These metrics quantify the proportion of predictions that match the expected results, providing a direct and essential measure of the model's ability to arrive at correct outputs under different reasoning paradigms.

**Accuracy.** It's the most widely used and task-agnostic metric for evaluating implicit reasoning performance (Liu et al., 2024c; Xu et al., 2025b;a), and measures whether the model produces the correct final answer, providing a coarse but robust signal of task success. Formally, for $N$ evaluation samples, it is defined as:

$$\text{Accuracy} = \frac{1}{N} \sum_{i=1}^{N} \mathbf{1} \left[ a_{\text{pred}}^{(i)} = a_{\text{gt}}^{(i)} \right], \tag{7}$$

where $a_{\text{pred}}^{(i)}$ is the model's predicted answer for the $i$-th instance, and $a_{\text{gt}}^{(i)}$ is the ground-truth answer.

**Pass@k, Pass@1.** It assesses the proportion of obtaining the correct answer at least once in $k$ independent outputs, usually used for code generation and mathematical reasoning tasks (Zhang et al., 2025c; Geiping et al., 2025; Kong et al., 2025). Rigorous Pass@1 denotes the proportion of directly obtaining correct answers in a single output and reduces to standard accuracy. Pass@k (Chen et al., 2021) can be formulated as:

$$\text{Pass@k} = 1 - \frac{\binom{n-c}{k}}{\binom{n}{k}}, \quad \text{Pass@1} = \frac{c}{n}, \tag{8}$$

where $n$ is the total number of samples and $c$ is the number of correct samples.

**Exact Match (EM).** It's a strict binary metric that requires the character-level match between the generated answer and the reference. If there is an exact match, the score will be 1, following the same form as Equation (7). This metric is suitable to evaluate tasks with deterministic answers, such as symbolic and mathematical reasoning (Cheng & Van Durme, 2024; Deng et al., 2023; 2024; Yue et al., 2025b).

**BLEU, ROUGE.** Both are widely used text-overlap metrics based on n-gram, designed to measure the similarity between generated text and reference texts. While originally designed for machine translation and summarization tasks, these metrics can also be applied to assess implicit reasoning by quantifying how closely the model's outputs align with expected answers or reasoning patterns, particularly in open-ended reasoning tasks where multiple valid answers may exist and exact string matching proves insufficient for comprehensive evaluation (Sun et al., 2025; Shen et al., 2025a; Goyal et al., 2024). BLEU focuses on n-gram precision with a brevity penalty which discourages overly short outputs, evaluating how much of the generated text appears in the reference content. ROUGE emphasizes recall, evaluating how much of the reference content appears in the generated text. Its most common forms are ROUGE-N (Goyal et al., 2024) and ROUGE-L (Sun et al., 2025; Shen et al., 2025a), which measure n-gram recall and compute the longest common subsequence, respectively.

Beyond these commonly used metrics, some studies have also employed METEOR (Shen et al., 2025a), preference accuracy (Gong et al., 2025) and BERTScore (Shen et al., 2025a) metrics to evaluate implicit reasoning performance, providing additional dimensions of assessment, such as semantic similarity.

### 5.1.2 Resource Efficiency

One of the core motivations behind implicit reasoning is its potential to reduce resource overhead by avoiding the explicit generation of intermediate steps, such as chain-of-thought sequences or reasoning traces.

Efficiency-oriented evaluation thus plays a crucial role in comparing implicit and explicit methods, particularly in resource-constrained or latency-sensitive settings.

Implicit reasoning methods commonly report the following metrics to evaluate efficiency:

**Decoding Latency.** It reports the *inference time* required to generate a complete response (Zhang et al., 2025a; Hao et al., 2024; Deng et al., 2024), usually including the time for the forward pass and decoding process. This metric can clearly reflect the low-latency advantage of implicit reasoning.

**Output Length.** It usually reports *the number of generated tokens* for the correct answer (Su et al., 2025; Shen et al., 2025a; Ma et al., 2025), particularly relevant for comparing implicit and explicit reasoning. Implicit reasoning generates fewer tokens due to internalizing the reasoning process.

**Computational Usage.** It usually reports *GPU usage* or *FLOPs* (Kong et al., 2025; Wang et al., 2025c), reflecting the demand on hardware resources. These two metrics are particularly important for evaluating implicit reasoning that introduces dynamic computation paths while maintaining low resource overhead.

**Accuracy per Computation Unit (ACU).** To evaluate the trade-off between reasoning performance and model efficiency, *CoT-Valve* (Ma et al., 2025) proposes a new metric called *Accuracy per Computation Unit (ACU)*. It quantifies how much accuracy a model achieves per unit of computational cost:

$$\text{ACU} = \frac{\text{Accuracy}}{\#\text{Params} \times \#\text{Tokens}}, \tag{9}$$

where #Params is the number of model parameters and #Tokens refers to the number of tokens generated in the reasoning process. This metric provides a unified view of model performance and computational cost.

Notably, some implicit approaches (e.g., token-wise depth adaptation or latent recurrence) introduce dynamic computation paths, making these metrics insufficient. In such cases, measuring adaptive depth or recurrence (Geiping et al., 2025; Chen et al., 2025c; Mohtashami et al., 2025; Saunshi et al., 2025) becomes necessary for a fair comparison of resource utilization.

### 5.1.3 Perplexity

Perplexity (PPL) is a fundamental metric for evaluating language modeling performance, quantifying the model's uncertainty when predicting the next token in a sequence and reflecting the model's ability to capture the statistical structure of language. A lower perplexity indicates that the model assigns higher probability to the correct token sequence.

Formally, it is defined as the exponential of the average negative log-likelihood over the evaluation corpus:

$$\text{PPL} = \exp\left(-\frac{1}{N}\sum_{i=1}^{N}\log p_\theta(w_i \mid w_{<i})\right) \tag{10}$$

where $N$ is the number of tokens in the evaluation corpus, $w_i$ denotes the $i$-th token, and $p_\theta(w_i \mid w_{<i})$ is the model's predicted probability of $w_i$ given its preceding context $w_{<i}$.

Some methods (Tack et al., 2025; Kong et al., 2025; Herel & Mikolov, 2024) combine perplexity with reasoning-oriented metrics to comprehensively evaluate the performance of implicit reasoning. Intuitively, strong language modeling capability serves as the foundation for effective reasoning abilities. Moreover, zero-shot perplexity evaluation can reflect whether a model has generalization ability, to some extent indicating implicit reasoning beyond mere memorization.

### 5.1.4 Probing Accuracy

Although implicit reasoning doesn't explicitly produce intermediate steps, relevant reasoning computations are usually encoded within the model's hidden states (Kong et al., 2025; Pfau et al., 2024). Understanding whether the model truly performs such reasoning necessitates the examination of its internal computational

Table 7: General knowledge and commonsense benchmarks for evaluating implicit reasoning. (§5.2.1)

| Dataset | Domain | Task Type | #Samples | Notes | Open Source |
|---------|--------|-----------|----------|-------|-------------|
| **CommonsenseQA** (Talmor et al., 2019) | Commonsense | Multiple choice QA | 12,247 | Five options | HuggingFace |
| **Social IQA** (Sap et al., 2019) | Social commonsense | Multiple choice QA | 37,588 | Three options | HomePage |
| **PIQA** (Bisk et al., 2020) | Physical commonsense | Multiple choice QA | 21K | 16K for training, 2K for development, 3K for testing | HuggingFace |
| **WinoGrande** (Sakaguchi et al., 2020) | Pronoun coreference | Fill-in-the-blank | 44K | Fill-in-the-blank format with two options | HuggingFace, GitHub |
| **HellaSwag** (Zellers et al., 2019) | Commonsense | Sentence continuation | 70K | Completing the sentence from four options | HuggingFace, HomePage |
| **SciQ** (Welbl et al., 2017) | Elementary, college-entry science exam | Multiple Choice QA | 13,679 | Science exam problems with four options | HuggingFace |
| **ARC-easy** (Clark et al., 2018) | US elementary, middle-school science | Multiple choice QA | 5,197 | Four options | HuggingFace |
| **ARC-challenge** (Clark et al., 2018) | US elementary, middle-school science | Multiple choice QA | 2,590 | Four options | HuggingFace |
| **TruthfulQA** (Lin et al., 2022) | General facts | Open-ended QA | 817 | Providing both generation and multiple-choice evaluation formats | HuggingFace, GitHub |

processes. Probing accuracy quantifies this by training auxiliary classifiers to predict intermediate labels from hidden representations (Brinkmann et al., 2024; Hou et al., 2023).

Let $h \in \mathbb{R}^d$ denote the hidden representation at a particular layer, $z$ denote an intermediate target (e.g., sub-result or logical step), and $N$ denote the number of samples. A linear transformation $f_\phi : \mathbb{R}^d \to \mathcal{Z}$ is trained to minimize the empirical risk:

$$\mathcal{L}_{\text{probe}} = \frac{1}{N} \sum_{i=1}^{N} \ell \left( f_\phi(h^{(i)}), z^{(i)} \right), \tag{11}$$

where $\ell(\cdot)$ is typically the cross-entropy loss for classification. Probing accuracy is then defined as:

$$\text{ProbingAcc} = \frac{1}{N} \sum_{i=1}^{N} \mathbf{1} \left[ f_\phi(h^{(i)}) = z^{(i)} \right]. \tag{12}$$

Probing serves as an indirect signal of internal reasoning processes, and successful probing reveals that models may indeed perform structured multi-step computations internally, even without explicit step-by-step outputs (Yu, 2024; Kudo et al., 2024). In practice, probing metrics can be complemented with causal or intervention-based analyses to enhance interpretability.

## 5.2 Benchmarks

Through a systematic analysis of widely used datasets in implicit reasoning, we organize these datasets into five primary categories and present a detailed review of each category in the following sections, highlighting their distinctive characteristics, representative datasets, and pivotal roles in advancing implicit reasoning evaluation. These benchmarks provide researchers with clear guidance for selecting appropriate evaluation instruments and conducting meaningful performance comparisons across different approaches.

### 5.2.1 General Knowledge and Commonsense Reasoning Benchmarks

Commonsense reasoning evaluates human-like cognitive abilities of models, requiring models to leverage everyday knowledge that humans typically take for granted. As summarized in Table 7, the following datasets assess whether the models can make intuitive inferences about physical commonsense, science knowledge, social interactions of humans, and everyday scenarios, effectively measuring their abilities of implicit reasoning on general knowledge. The following introduces the characteristics of each dataset.

- **CommonsenseQA** (Talmor et al., 2019): A benchmark designed to evaluate the ability of models to draw upon commonsense understanding, rather than relying solely on explicit factual information.

- **Social IQA** (Sap et al., 2019): The dataset requires models to reason about people's motivations, emotions, and likely reactions, evaluating models' understanding of social interactions and human behavior in everyday situations.

- **PIQA** (Bisk et al., 2020): A dataset designed to evaluate commonsense reasoning about physical interactions like physical phenomena, properties, and manipulations, requiring models to select the most appropriate solution from two given alternatives.

- **WinoGrande** (Sakaguchi et al., 2020): An adversarial Winograd Schema Challenge dataset at scale for commonsense reasoning. It requires the models to select the correct option to fill in the blanks, and this selection often involves understanding the referential relationship of pronouns in the sentence.

- **HellaSwag** (Zellers et al., 2019): A dataset for commonsense natural language inference, employing adversarial filtering to generate challenging distractors. It requires models to select the most plausible continuation from four options given a context describing everyday activities.

- **SciQ** (Welbl et al., 2017): It collects 13.7K science exam questions covering biology, chemistry, earth science and physics from elementary to college-entry level. Each question typically includes four answer options and a paragraph of supporting evidence for the correct answer.

- **ARC-easy** (Clark et al., 2018), **ARC-challenge** (Clark et al., 2018): ARC dataset extracts 7,787 problems from 3-grade to 9-grade science across 80 science topics. It is partitioned into two subsets: an Easy set of 5197 questions and a Challenge set of 2590 difficult questions.

- **TruthfulQA** (Lin et al., 2022): A benchmark designed to evaluate the truthfulness of language models' responses across 38 categories, testing whether models can avoid generating false answers learned from human falsehoods.

### 5.2.2 Mathematical Reasoning and Programming Benchmarks

Mathematical and programming reasoning benchmarks are essential for comprehensively evaluating models' abilities to solve complex, structured problems that require symbolic manipulation, algorithmic thinking and multi-step reasoning. As shown in Table 8, these datasets span a wide range of difficulties, from elementary arithmetic to competitive programming, evaluating the internalization of mathematical knowledge and computational thinking.

- **GSM8K** (Cobbe et al., 2021), **GSM-Hard** (Gao et al., 2023), **GSM8K-Aug** (Deng et al., 2023): GSM8K is a standard benchmark for grade-school math problems, often used to test multi-step numerical reasoning. Many implicit methods report performance on this dataset to compare against CoT-based baselines. GSM-Hard is derived from GSM8K dataset by replacing original numbers with random larger integers. And GSM8K-Aug is generated using GPT-4, based on the GSM8K training set.

- **SVAMP** (Patel et al., 2021): As simple variations on arithmetic math word problems, SVAMP is created by applying certain variations to existing datasets like ASDiv dataset.

- **AQUA-RAT** (Ling et al., 2017): This provides multiple-choice algebra problems, requiring models to both select correct answers and generate natural language explanations for the reasoning process.

- **MATH** (Hendrycks et al., 2021b), **MATH-500** (Lightman et al., 2024): This is sourced from high school math competitions such as AMC 10, AMC 12 and AIME (AIM), with each problem accompanied by complete step-by-step solutions. Lightman et al. (2024) create the MATH-500 dataset by selecting 500 test problems uniformly at random from the test problems of MATH dataset to avoid over-fitting on the MATH training problems.

Table 8: Mathematical reasoning and programming benchmarks for evaluating implicit reasoning. (§5.2.2)

| Dataset | Domain | Task Type | #Samples | Notes | Open Source |
|---|---|---|---|---|---|
| **GSM8K** (Cobbe et al., 2021) | Grade-school math | Math word problems | 8.5K | Needing 2 and 8 steps to solve problems | HuggingFace |
| **GSM-Hard** (Gao et al., 2023) | Grade-school math | Math word problems | 1.32K | Harder variants of GSM8K | HuggingFace |
| **GSM8K-Aug** (Deng et al., 2023) | Grade-school math | Math word problems | 378K | Generated by GPT-4 | GitHub |
| **SVAMP** (Patel et al., 2021) | Grade-school math | Math word problems | 1,000 | Focusing on math of grades four and lower | GitHub |
| **AQUA-RAT** (Ling et al., 2017) | Algebraic math | Multiple choice math QA | 100K | Algebra word problems with five options and rationale descriptions | HuggingFace, GitHub |
| **MATH** (Hendrycks et al., 2021b) | High-school competition math | Mathematical reasoning QA | 12,500 | Step-by-step solutions written in LATEX | GitHub |
| **MATH-500** (Lightman et al., 2024) | High-school competition math | Mathematical reasoning QA | 500 | Selected from MATH dataset uniformly at random | HuggingFace, GitHub |
| **CollegeMath** (Tang et al., 2024) | College math | Math word problems | 4,099 | Needing logical and mathematical reasoning | GitHub |
| **Fresh-Gaokao-Math-2023** (Tang et al., 2024) | High-school math | Math word problems | 30 | 30 questions from Gaokao math exam | GitHub |
| **AIME** (AIM) | Competition-level math | Mathematical reasoning QA | – | Samples from American Invitational Mathematics Examination | HomePage |
| **MultiArith** (Roy & Roth, 2015) | Grade-school math | Arithmetic word problems | 600 | Multi-step arithmetic problems | HuggingFace |
| **ASDiv** (Miao et al., 2020) | Grade-school math | Math word problems | 2,305 | XML format | GitHub |
| **ASDiv-Aug** (Xu et al., 2025b) | Grade-school math | Math word problems | 5,221 | Augmented from ASDiv dataset | HuggingFace |
| **MAWPS** (Koncel-Kedziorski et al., 2016) | Arithmetic and algebraic math | Math Word Problems | 2,373 | Integration of existing datasets | GitHub |
| **MathQA** (Amini et al., 2019) | Algebraic math | Multiple Choice QA | 37K | Re-annotating AQUA-RAT dataset | HuggingFace, HomePage |
| **Mathematics Dataset** (Saxton et al., 2019) | Algebra, calculus, polynomials, probability, etc. | Mathematical reasoning QA | 2M | Free-form textual input/output format | HuggingFace, GitHub |
| **HumanEval** (Chen et al., 2021) | Programming | Code Generation | 164 | Manually-crafted Python problems | HuggingFace, GitHub |
| **MBPP** (Austin et al., 2021) | Programming | Code Generation | 974 | Python problems with task description, function signature, and assert-based test cases | HuggingFace, GitHub |
| **LiveCodeBench** (Jain et al., 2025) | Programming | Code Generation | 880 | Dynamic, contamination-free code benchmark | HuggingFace, HomePage |

- **CollegeMath** (Tang et al., 2024), **Mathematics Dataset** (Saxton et al., 2019): These datasets focus on complex mathematics problems, including calculus, linear algebra, and statistics, filling the evaluation gap between elementary arithmetic and advanced mathematics and providing a comprehensive assessment of models' abilities to handle complex mathematics.

- **Fresh-Gaokao-Math-2023** (Tang et al., 2024): To avoid potential data contamination and evaluation bias, Tang et al. (2024) construct the Fresh-Gaokao-Math-2023 dataset from 2023 Chinese Gaokao mathematics examination, thereby enabling reliable evaluation of reasoning ability rather than memorization effects.

- **AIME** (AIM): The dataset comprises challenging problem sets from multiple years of the American Invitational Mathematics Examination, testing advanced mathematical reasoning of models. In particular, AIME24 (2024 exam) (MathAI, 2024a) and AIME25 (2025 exam) (MathAI, 2024b) have become the most widely adopted benchmarks for recent LLM evaluations.

- **MultiArith** (Roy & Roth, 2015): A dataset for evaluating multi-step arithmetic reasoning involving basic operations (addition, subtraction, multiplication, division).

- **ASDiv** (Miao et al., 2020), **ASDiv-Aug** (Xu et al., 2025b): The ASDiv dataset focuses on arithmetic and algebraic word problems to evaluate the mathematical reasoning. Xu et al. (2025b) create a hard version of ASDiv, named ASDiv-Aug, through instance replication and randomly numerical substitution, evaluating LLMs' reasoning abilities rather than memory capacity.

- **MAWPS** (Koncel-Kedziorski et al., 2016): A unified dataset integrated from several existing collections, including AddSub (Hosseini et al., 2014), SingleOp (Roy et al., 2015), MultiArith (Roy & Roth, 2015), SingleEq (Koncel-Kedziorski et al., 2015), and SimulEq (Kushman et al., 2014).

- **HumanEval** (Chen et al., 2021), **MBPP** (Austin et al., 2021), **LiveCodeBench** (Jain et al., 2025): Three representative programming benchmarks. HumanEval comprises manually-crafted programming problems and evaluates functional correctness through unit tests. MBPP focuses on entry-level programming tasks, testing fundamental coding skills and standard library usage. LiveCodeBench is a dynamic and contamination-free benchmark by continuously sourcing problems from competitive coding platforms, assessing the capabilities of code generation, self-repair, code execution, and test output prediction. As of January 2025, LiveCodeBench has released 880 problems. These datasets evaluate models on language comprehension, reasoning, algorithms, and mathematics thinking.

### 5.2.3 Language Modeling and Reading Comprehension Benchmarks

Language modeling and reading comprehension benchmarks evaluate fundamental capabilities of language models such as long-range dependency modeling, contextual inference, and semantic understanding, building the foundation for evaluating a variety of reasoning capabilities. As shown in Table 9, these benchmarks span from basic word prediction to complex reasoning-intensive language comprehension tasks requiring numerical and logical reasoning, measuring models' capacity to process and reason over textual information.

- **Penn Treebank (PTB)** (Marcus et al., 1993): A richly annotated corpus of English text from Wall Street Journal, originally used for syntactic and semantic analysis. It offers detailed syntactic structures, which are essential for training and evaluating models in natural language processing tasks such as parsing and language modeling.

- **One Billion Word Benchmark** (Chelba et al., 2013): A large-scale language modeling corpus derived from WMT 2011 News Crawl data. This benchmark encompasses approximately 0.8 billion words of textual content and has become a standard benchmark for evaluating language models on large-vocabulary prediction tasks.

- **WikiText-2** (Merity et al., 2017), **WikiText-103** (Merity et al., 2017): WikiText dataset encompasses approximately 103 million tokens sourced from high-quality Wikipedia articles, including WikiText-2 and WikiText-103, whose vocabularies have different sizes. WikiText-2 dataset is approximately twice the size of PTB while WikiText-103 is approximately 110 times larger than PTB, providing a more substantial corpus for evaluating long-term dependency and language modeling.

- **LAMBADA** (Paperno et al., 2016): It requires models to predict the masked word based on the broader narrative contexts, assessing the capabilities of long-range context comprehension and language modeling.

- **AG News** (Zhang et al., 2015): It comprises a large collection of news articles across four topical categories (i.e., World, Sports, Business, and Science/Technology), and is originally designed for text classification. Recently, Kong et al. (2025) also use AG News for zero-shot unconditional perplexity evaluation, showing its applicability to zero-shot language modeling.

- **PubMed** (Cohan et al., 2018), **arXiv** (Cohan et al., 2018): These large-scale corpora consist of scientific papers sourced from scholarly repositories PubMed and arXiv, providing structured long documents with abstracts as ground-truth summaries for abstractive summarization.

- **WikiSplit** (Botha et al., 2018): It encompasses one million sentences extracted from Wikipedia's revision history, supports split-and-rephrase tasks by providing sentence pairs where complex sentences are decomposed into simpler and shorter ones while preserving semantic equivalence.

- **SQuAD** (Rajpurkar et al., 2016), **SQuAD 2.0** (Rajpurkar et al., 2018): Both are large-scale benchmarks for reading comprehension and question-answering. SQuAD 2.0 extends SQuAD by introducing over 50K adversarially constructed unanswerable questions, further challenging models to recognize the unanswerability of questions and abstain from answering when appropriate.

Table 9: Language modeling and reading comprehension benchmarks for implicit reasoning in LLMs. (§5.2.3)

| Dataset | Category | Task Type | #Samples | Source | Notes | Open Source |
|---------|----------|-----------|----------|--------|-------|-------------|
| **Penn Treebank (PTB)** (Marcus et al., 1993) | Language modeling | Next word prediction | – | Wall Street Journal text | 929K training tokens, 73K validation tokens, 82K test tokens | HomePage |
| **One Billion Word Benchmark** (Chelba et al., 2013) | Language modeling | Next word prediction | – | News | Nearly 1 billion words | GitHub |
| **WikiText-2** (Merity et al., 2017) | Language modeling | Next word prediction | 44.8K | Wikipedia | Over 2 million tokens | HuggingFace |
| **WikiText-103** (Merity et al., 2017) | Language modeling | Next word prediction | 1.81M | Wikipedia | Over 103 million tokens | HuggingFace |
| **LAMBADA** (Paperno et al., 2016) | Language modeling | Cloze-style word prediction | 10,022 | Narrative novels | Requiring long-range context | HuggingFace, HomePage |
| **AG News** (Zhang et al., 2015) | Language modeling | Text classification | 127.6K | News | Four classes in total | HuggingFace |
| **PubMed** (Cohan et al., 2018) | Language modeling | Abstractive summarization | 133K | Scientific papers | Long and structured document summarization | HuggingFace, GitHub |
| **arXiv** (Cohan et al., 2018) | Language modeling | Abstractive summarization | 215K | Scientific papers | Long and structured document summarization | HuggingFace, GitHub |
| **WikiSplit** (Botha et al., 2018) | Language modeling | Split-and-rephrase | 1M | Wikipedia | Splitting complex sentences into simpler ones | HuggingFace, GitHub |
| **SQuAD** (Rajpurkar et al., 2016) | Reading comprehension | Extractive QA | 100K | Wikipedia | Span-based answers | HuggingFace, HomePage |
| **SQuAD 2.0** (Rajpurkar et al., 2018) | Reading comprehension | Extractive QA | 150K | Wikipedia | Adding 50K unanswerable questions | HuggingFace, HomePage |
| **QuAC** (Choi et al., 2018) | Dialogue comprehension | Open-ended QA | 98,407 | Wikipedia | 14K dialogs | HuggingFace, HomePage |
| **CoQA** (Reddy et al., 2019) | Dialogue comprehension | Extractive QA | 127K | Multi-domain knowledge | 8K dialogs; 7 domains | HuggingFace, HomePage |
| **BoolQ** (Clark et al., 2019) | Reading Comprehension | Boolean QA | 12,697 | Wikipedia | Return "yes" or "no" as output | HuggingFace, GitHub |
| **DROP** (Dua et al., 2019) | Reading comprehension | Extractive QA | 96,567 | Wikipedia | Discrete reasoning (such as addition, counting, or sorting) over text | HuggingFace, HomePage |
| **NarrativeQA** (Kočiský et al., 2018) | Reading comprehension | Abstractive QA | 46,765 | Books, movie scripts | Requiring reading entire contents for answering | HuggingFace |
| **RACE** (Lai et al., 2017) | Reading comprehension | Multiple choice QA | 97,687 | Chinese middle-, high-school English exams | Four options; including RACE-M with 28,293 questions, RACE-H with 69,394 questions | HuggingFace, HomePage |
| **Natural Questions** (Kwiatkowski et al., 2019) | Reading comprehension | Open-ended QA | 323K | Wikipedia | Questions from real Google queries with Wikipedia-based answers | HuggingFace, HomePage |
| **WebQuestions** (Berant et al., 2013) | Reading comprehension | Open-ended QA | 5,810 | Freebase | Questions from Google Suggest API with Freebase-based answers | HuggingFace |
| **TriviaQA** (Joshi et al., 2017) | Reading comprehension | Open-ended QA | 95K | Trivia | 650K question-answer-evidence triples | HuggingFace, HomePage |
| **TydiQA** (Clark et al., 2020) | Reading comprehension | Extractive QA | 204K | Wikipedia | 11 typologically diverse languages | HuggingFace |
| **LogiQA** (Liu et al., 2021) | Reading Comprehension | Multiple Choice QA | 8,678 | National Civil Servants Examination of China | Four options; logical deductive reasoning | HuggingFace, GitHub |

- **QuAC** (Choi et al., 2018), **CoQA** (Reddy et al., 2019): Both are large-scale conversational benchmarks for dialogue-based reading comprehension, facilitating multi-turn dialogue comprehension and contextual reasoning. QuAC simulates information-seeking dialogue scenarios where students raise open-ended questions from unseen Wikipedia articles and teachers answer the questions based on Wikipedia content. CoQA is extracted from approximately 8K dialogues spanning 7 distinct domains, allowing free-form answers with text spans as rationales.

- **BoolQ** (Clark et al., 2019): It consists of questions, Wikipedia paragraphs, and binary Yes/No answers. This benchmark needs sophisticated entailment reasoning and non-factual inference, rather than straightforward factual retrieval.

- **DROP** (Dua et al., 2019): The dataset requires models to perform discrete reasoning over paragraphs such as counting, comparison, addition, and subtraction. DROP enhances reading comprehension beyond simple text matching by incorporating numerical reasoning.

- **NarrativeQA** (Kočiský et al., 2018): It is designed for evaluating the ability of models to understand and summarize long-form narratives from books and movie scripts, and requires models to read the entire story content, encouraging deeper comprehension of language.

- **RACE** (Lai et al., 2017): It is derived from English exams for Chinese students, encompassing 97,687 questions in total. It can be divided into two subsets: the RACE-M from middle-school English exams and the RACE-H from high-school English exams. It usually requires more reasoning, such as passage summarization and attitude analysis to gain better performance.

- **Natural Questions**(Kwiatkowski et al., 2019), **WebQuestions**(Berant et al., 2013): Both are constructed from Google search queries. Natural Questions pairs queries with corresponding entire Wikipedia pages, requiring models to understand the full Wikipedia article for answers. WebQuestions links queries from Google Suggest API to Freebase answers without relying on expert-annotated logical forms, enabling large-scale QA over structured knowledge graphs.

- **TriviaQA** (Joshi et al., 2017): It encompasses 95K trivia-related QA pairs and provides 650K question-answer-evidence triples where evidence documents are collected from Wikipedia and the Web search. TriviaQA has more compositional questions and needs more cross-sentence reasoning.

- **TydiQA** (Clark et al., 2020): A multilingual corpus for reading comprehension across 11 typologically diverse languages, including Arabic, Bengali, and Russian. Unlike monolingual datasets, TydiQA emphasizes cross-lingual generalization and linguistic diversity, requiring models to handle diverse linguistic features inherent to each language without the use of translation.

- **LogiQA** (Liu et al., 2021): A large-scale logical reasoning dataset for reading comprehension across 5 types of logical reasoning (i.e., categorical reasoning, sufficient conditional reasoning, necessary conditional reasoning, disjunctive reasoning and conjunctive reasoning), evaluating the depth and robustness of models' understanding and reasoning abilities.

### 5.2.4 Complex Multi-hop and Multidisciplinary QA Benchmarks

Multi-hop and multidisciplinary QA benchmarks are well-suited for evaluating implicit reasoning capabilities that involve decomposing complex questions, retrieving relevant evidence, and performing latent multi-step reasoning. These benchmarks require models to construct reasoning chains across multiple knowledge sources or domains, making them essential for assessing reasoning depth and generalization. Table 10 presents comprehensive statistics of these datasets.

- **HotpotQA** (Yang et al., 2018), **2WikiMultiHopQA** (Ho et al., 2020): Both are derived from Wikipedia to evaluate multi-hop reasoning capabilities. HotpotQA requires models to aggregate and synthesize information from multiple supporting documents to derive answers, and provides sentence-level supporting facts for explainable prediction. 2WikiMultiHopQA combines unstructured Wikipedia and structured Wikidata, introducing evidence information for explainable prediction.

- **ProntoQA** (Saparov & He, 2023), **ProsQA** (Hao et al., 2024): A synthetic question-answering dataset to evaluate logical reasoning capabilities. ProntoQA randomly generates tree-structured ontologies, derives proofs from ontologies, and then converts them into natural language to construct examples with True/False answers. This dataset enables researchers to parse the models' reasoning chains into symbolic proofs for formal analysis. ProsQA is constructed from randomly generated directed acyclic graphs (DAGs), demanding more sophisticated planning and search strategies.

- **StrategyQA** (Geva et al., 2021): The dataset challenges models to perform implicit multi-step reasoning to arrive at yes/no answers, requiring models to decompose complex questions into multi-step strategies. Each question consists of decomposed steps and corresponding evidence paragraphs from Wikipedia for each step.

- **ComplexWebQuestions** (Talmor & Berant, 2018): Designed to evaluate multi-hop reasoning capabilities, the dataset features complex compositional questions, which require decomposition into

Table 10: Multi-hop and multidisciplinary QA benchmarks for evaluating implicit reasoning. (§5.2.4)

| Dataset | Category | Task Type | #Samples | Source | Notes | Open Source |
|---|---|---|---|---|---|---|
| **HotpotQA** (Yang et al., 2018) | Multi-hop QA | Open-ended QA | 113K | Wikipedia | Multi-document QA with sentence-level supporting facts | HuggingFace, HomePage |
| **2WikiMultiHopQA** (Ho et al., 2020) | Multi-hop QA | Open-ended QA | 192,606 | Wikipedia | From unstructured Wikipedia and structured Wikidata data, with evidence information for interpretability | GitHub |
| **Bamboogle** (Press et al., 2023) | Multi-hop QA | Open-ended QA | 125 | Wikipedia | Manually writing difficult 2-hop questions from Wikipedia articles | HuggingFace, GitHub |
| **ProntoQA** (Saparov & He, 2023) | Multi-hop QA | Logical reasoning | 10K | Ontologies | synthetic dataset with True/False answers | GitHub |
| **ProsQA** (Hao et al., 2024) | Multi-hop QA | Logical reasoning | 18,686 | Directed acyclic graphs | synthetic dataset with directed acyclic graphs | GitHub |
| **ComplexWebQuestions** (Talmor & Berant, 2018) | Multi-hop QA | Open-ended QA | 34,689 | Web snippets, Freebase | Web-based complex QA with question decomposition | HuggingFace |
| **OpenBookQA** (Mihaylov et al., 2018) | Multi-hop QA | Multiple Choice QA | 5,957 | Elementary-level science facts | Requiring combining science facts with multi-hop reasoning | HuggingFace, HomePage |
| **StrategyQA** (Geva et al., 2021) | Multi-hop QA | Boolean QA | 2,780 | Wikipedia | Focusing on implicit multi-hop reasoning | GitHub |
| **BIG-Bench** (Srivastava et al., 2023) | Multidisciplinary | Multiple Choice QA, Math word problems, etc | – | Math, biology, physics, commonsense, etc | 204 tasks in total | HuggingFace, GitHub |
| **BIG-Bench Hard (BBH)** (Suzgun et al., 2023) | Multidisciplinary | Multiple Choice QA, Math word problems, etc | – | Math, commonsense, etc | 23 challenging tasks from BIG-Bench | HuggingFace, GitHub |
| **MMLU** (Hendrycks et al., 2021a) | Multidisciplinary | Multiple Choice QA | 15,908 | STEM, humanities, social sciences, etc | 57 tasks across STEM, humanities, social science, etc.; four options | HuggingFace, GitHub |
| **GPQA** (Rein et al., 2024) | Multidisciplinary | Multiple choice QA | 448 | Graduate-level physics, chemistry, and biology | High-difficulty questions with four options written by domain experts; Graduate-level physics, chemistry, and biology | HuggingFace, GitHub |

simpler sub-questions and reasoning across multiple web snippets. Each example in ComplexWebQuestions consists of a question, an answer, a SPARQL query for Freebase, and relevant web snippets, enabling interaction with the web, reading comprehension, and semantic parsing.

- **OpenBookQA** (Mihaylov et al., 2018): This dataset provides elementary science facts as an open book reference, and requires not only the combination of commonsense knowledge with scientific facts, but also multi-hop reasoning abilities.

- **Bamboogle** (Press et al., 2023): Designed to investigate the compositional reasoning abilities, it consists of 2-hop difficult questions that internet search engines can't answer correctly, with supporting evidence from Wikipedia.

- **BIG-Bench** (Srivastava et al., 2023): A large-scale collaborative benchmark comprising 204 diverse tasks, such as date understanding, question answering, multi-digit multiplication. It is designed to probe the capabilities and limitations of language models across reasoning, mathematics, understanding domains, etc.

- **BIG-Bench Hard (BBH)** (Suzgun et al., 2023): A subset of 23 challenging tasks selected from BIG-Bench, specifically focusing on tasks where language model evaluations failed to outperform human-raters. These tasks require multi-step reasoning and represent the most difficult challenges from BIG-Bench.

- **MMLU** (Hendrycks et al., 2021a): A comprehensive evaluation benchmark designed to assess language models' proficiency in world knowledge and problem-solving ability. This benchmark comprises 15,908 multiple-choice questions spanning 57 subjects across STEM (i.e., science, technology, engineering, and mathematics), the humanities, the social sciences, etc.

- **GPQA** (Rein et al., 2024): It is written by domain experts in physics, chemistry, and biology, establishing a high-quality benchmark for evaluating LLMs' reasoning capabilities. Even with internet access, non-experts often struggle to answer these questions correctly. The dataset is released in three versions: the main set (448 questions), the extended set (546 questions), and the diamond set (198 questions), with the diamond version being the most challenging.

Table 11: Multi-modal benchmarks for evaluating implicit reasoning in LLMs. (§5.2.5)

| Dataset | Modality | Task Type | #Samples | Source | Notes | Open Source |
|---|---|---|---|---|---|---|
| **LLaVA-CoT-100K** (Xu et al., 2024) | Text, image | Visual QA | 100K | Multi-domain knowledge | Providing detailed structured reasoning annotations generated by GPT-4o | HuggingFace, GitHub |
| **MMStar** (Chen et al., 2024b) | Text, image | Visual multiple choice QA | 1,500 | Multi-domain knowledge | Covering logical reasoning, mathematics, instance reasoning and etc. | HuggingFace, HomePage |
| **MMBench** (Liu et al., 2024d) | Text, image | Visual multiple choice QA | 3,217 | Multi-domain knowledge | Bilingual benchmark of Chinese and English | GitHub |
| **MM-Vet** (Yu et al., 2024c) | Text, image | Visual open-ended QA | 218 | Multi-domain knowledge | 200 images in total, including 3 medical images | GitHub |
| **AI2D-RST** (Hiippala et al., 2021) | Text, image | Visual QA | 1K | Primary school natural science diagram | Using Rhetorical Structure Theory (RST) | GitHub |
| **HallusionBench** (Guan et al., 2024) | Text, image, video | Visual QA | 1,129 | Multi-domain knowledge | Diagnosing hallucination and illusion of VLMs | GitHub |
| **MathVista** (Lu et al., 2024) | Text, image | Visual QA | 6,141 | Math, science | Creating 3 new datasets (i.e., IQTest, FunctionQA, PaperQA) | HuggingFace, HomePage |
| **OlympiadBench-Math** (He et al., 2024) | Text, image | Visual QA | 6,142 | Olympic math competition | 2,911 questions with image input | HuggingFace, GitHub |
| **ScienceQA** (Lu et al., 2022) | Text, image | Visual multiple choice QA | 21,208 | Elementary, high school science | Each problem with lectures and explanations | HuggingFace, HomePage |
| **TheoremQA** (Chen et al., 2023) | Text, image | Visual QA | 800 | University-level theorems | 51 questions with image input | HuggingFace |

### 5.2.5 Multi-modal Reasoning Benchmarks

Multimodal benchmarks can evaluate the implicit reasoning capabilities of vision-language models (VLMs) that go beyond unimodal textual understanding, as shown in Table 11. These datasets require VLMs to extract relevant visual features, establish correspondences with textual content, and synthesize multi-modal information to generate responses. Notably, many of these benchmarks incorporate mathematical diagrams and scientific figures that mirror real-world learning scenarios, making them particularly valuable for assessing models' potential in educational applications and automated tutoring systems.

- **LLaVA-CoT-100K** (Xu et al., 2024): Designed to enhance the multistage reasoning capabilities of VLMs in reasoning-intensive tasks, this dataset is sampled from diverse general and science-focused VQA datasets, and annotated with structured reasoning processes generated by GPT-4o, including summary, caption, reasoning, and conclusion.

- **MMStar** (Chen et al., 2024b): It addresses key limitations of visual redundancy and data leakage in existing visual benchmarks, spanning 6 core capabilities and 18 fine-grained axes like logical reasoning, mathematics, and fine-grained perception. The benchmark also introduces Multi-modal Gain (MG) and Leakage (ML) metrics for precise performance evaluation.

- **MMBench** (Liu et al., 2024d): A bilingual benchmark for robust evaluation of VLMs. This benchmark spans 20 fine-grained ability dimensions including logical reasoning, social reasoning, fine-grained perception, etc. Furthermore, it introduces a novel circular evaluation strategy (i.e., CircularEval) for rigorous evaluation.

- **MM-Vet** (Yu et al., 2024c): Designed to evaluate the abilities of VLMs on complex tasks, it integrates 6 core vision-language capabilities, including recognition, OCR, knowledge, language generation, spatial awareness, and math, and evaluates 16 tasks, emphasizing real-world scenarios.

- **MathVista** (Lu et al., 2024): A mathematical reasoning benchmark in visual contexts, covering 7 mathematical reasoning types and 5 primary tasks. It comprises 6,141 examples, including 228 from the newly created IQTest dataset for logical reasoning, 400 from the newly created FunctionQA dataset for algebraic reasoning, 108 from the newly created PaperQA dataset for scientific reasoning, and 5,405 examples from 28 existing datasets (e.g., MathQA (Amini et al., 2019)).

- **AI2D-RST** (Hiippala et al., 2021): A multimodal corpus of 1,000 English-language diagrams from primary school natural sciences. AI2D-RST is built on the AI2D dataset (Kembhavi et al., 2016) by introducing a multi-layer annotation schema to capture perceptual grouping, connectivity, and discourse relations using Rhetorical Structure Theory (RST). And AI2D-RST uses graphs to represent annotation layers.

- **HallusionBench** (Guan et al., 2024): A diagnostic benchmark designed to evaluate language hallucination and visual illusion in large VLMs. It provides visual input modalities of image and video, evaluating visual commonsense and image-context reasoning.

- **ScienceQA** (Lu et al., 2022): A large-scale multimodal benchmark for science QA across elementary and high school science curricula. Each question is annotated with lectures (background knowledge) and explanations (reasoning chains), enabling research into interpretable reasoning.

- **TheoremQA** (Chen et al., 2023): A theorem-driven QA dataset collected from 350 theorems spanning mathematics, physics, EE&CS, and finance, evaluating models' abilities to apply mathematical and scientific theorems. Among them, 51 questions have image inputs.

- **OlympiadBench-Math** (He et al., 2024): This dataset is a subset of OlympiadBench (He et al., 2024), which is an Olympiad-level bilingual multimodal scientific benchmark, and contains 6,142 mathematics problems and 2,334 physics problems sourced from prestigious Olympiad-level mathematics and physics competitions.

## 6 Challenges and Future Directions

Despite growing interest and rapid progress, implicit reasoning in LLMs remains in its early stages. Current methods face several critical challenges that hinder their theoretical understanding, practical reliability, and wide-scale adoption. Below, we highlight six key limitations, each paired with promising research directions.

**Limited Interpretability and Latent Opacity.** Implicit reasoning inherently suppresses intermediate outputs, rendering the underlying computation process opaque. This lack of visibility prevents us from knowing whether the model is performing genuine multi-step reasoning or merely exploiting memorized knowledge and spurious correlations. Existing probing and attribution techniques offer only coarse, indirect insights into internal dynamics (Kudo et al., 2024; Zhang & Viteri, 2025; Lin et al., 2025), and our understanding of how implicit reasoning unfolds in latent space remains limited (Yang et al., 2025b). As architectures evolve toward purely latent reasoning models that never verbalize any steps (Hao et al., 2024; Geiping et al., 2025), these tools will lose even their indirect effectiveness. To reveal the hidden reasoning process, future work should develop finer-grained methods such as causal intervention analysis, state-trajectory visualization, and attribution approaches tailored to dynamic computation flows in implicit reasoning, alongside mechanistic studies that uncover structural patterns supporting implicit reasoning.

**Limited Control and Reliability.** In contrast to explicit prompting, implicit reasoning provides no built-in mechanisms for guiding, inspecting, or correcting the internal reasoning process. When latent computation fails, it often does so silently without emitting intermediate signals or uncertainty estimates, resulting in brittle behavior and reduced reliability in high-stakes applications (Kudo et al., 2024). Empirical studies have shown that prompted implicit reasoning can skip or fuse critical steps without warning, making the process difficult to monitor or steer (Liu et al., 2024b). Moreover, monitoring may alter model behavior, producing superficial reasoning traces while true computation remains hidden (Emmons et al., 2025; Arnav et al., 2025), further undermining reliability. Enhancing controllability thus calls for models to support adjustable reasoning budgets, confidence-aware execution, and reversible computation flows, as well as hybrid strategies that allow partial intervention, such as steering internal states or verifying latent transitions, to strike a practical balance between silent reasoning and robust supervision.

**Performance Gap Compared to Explicit Reasoning.** While implicit methods offer faster reasoning and better alignment with end-task formats, they often underperform explicit strategies such as Chain-of-Thought in terms of final accuracy (Deng et al., 2023; Hao et al., 2024). This performance gap arises partly

because implicit models tend to rely on shortcut heuristics rather than robust, generalizable reasoning (Lin et al., 2025). Comparative studies further show that even architectures with looped execution designed for supporting internal multi-step computation still fall short of explicit methods on complex compositional or open-ended tasks (Xu & Sato, 2025). To close this gap, future work needs to explore hybrid strategies that combine silent reasoning with lightweight verification or align training objectives more directly with latent reasoning fidelity.

**Lack of Standardized Evaluation.** Current evaluations focus almost exclusively on final-answer correctness, without assessing the quality, depth, or stability of internal reasoning. As a result, it remains difficult to diagnose failure modes or to distinguish genuine reasoning from shallow heuristics. Moreover, existing studies adopt highly inconsistent benchmarking practices, often using self-curated subsets or proprietary datasets despite the availability of widely-used benchmarks such as CommonsenseQA (Talmor et al., 2019), GSM8K (Cobbe et al., 2021) or HotpotQA (Yang et al., 2018). As we summarize in Section 5.2, over 70 datasets have been used in isolation across the literature, hindering fair comparability and reproducibility. There is a pressing need for unified benchmark suites tailored to implicit reasoning. Such benchmarks (Hagendorff & Fabi, 2025) should incorporate latent annotations, standardized probing protocols, and metrics that assess internal consistency, trajectory depth, and robustness to distributional shifts.

**Architecture and Generalization Constraints.** Many existing approaches to implicit reasoning depend on architecture-specific components, such as loop controllers (Saunshi et al., 2025; Mohtashami et al., 2025), planning tokens (Wang et al., 2024c), or task-specific latent heads (Xu et al., 2025b; Tan et al., 2025). Furthermore, implicit reasoning is usually evaluated on smaller-scale models (Deng et al., 2023; Orlicki, 2025). Such constraints are difficult to generalize across model families or efficiently scale to larger systems, such as poor compatibility with standard transformer architectures and the challenges for integration into pretraining workflows (Zeng et al., 2025; Ruan et al., 2025). Future work may benefit from architecture-agnostic designs and reasoning objectives that can naturally integrate into mainstream training pipelines and larger models, and should further explore the capabilities and performance of implicit reasoning on larger-parameter models.

**Dependence on Explicit Supervision.** Most current implicit reasoning methods are not trained purely in the latent space, but rely directly on explicit reasoning traces (e.g., Chain-of-Thought) to guide latent reasoning (Deng et al., 2023; Hao et al., 2024; Shen et al., 2025b). This dependence undermines the independence of implicit reasoning and restricts its scalability, since explicit annotations are costly and unavailable in many domains. Future research should investigate supervision signals that operate directly on latent trajectories, including self-consistency constraints, implicit verification objectives, or unsupervised discovery of latent reasoning structures.

# 7 Conclusion

This survey provides a comprehensive account of implicit reasoning in LLMs, where reasoning unfolds internally without explicit step-by-step traces. We distinguish implicit reasoning from explicit reasoning methods, and organize existing approaches into a structured taxonomy that captures the diversity of execution strategies. Alongside methodological review, we examine structural and behavioral evidence that supports the presence of such latent reasoning processes, and summarize prevailing evaluation practices.

Despite growing interest, implicit reasoning remains a developing paradigm with substantial open questions. By analyzing current limitations and highlighting unresolved challenges, this survey aims to clarify the landscape and inform future research directions in the pursuit of more efficient, robust, and cognitively aligned language models.

**Broader Impact Statement**

This survey provides a comprehensive review of implicit reasoning in LLMs, focusing on methodological insights, empirical evidence, and evaluation strategies. As a synthesis of existing research, it does not introduce

new models or datasets, nor does it involve deployment in real-world systems. Most of the studies covered are evaluated on general-purpose reasoning benchmarks in domains such as mathematics, commonsense, and question answering, without involving sensitive areas like finance, healthcare, or law. Therefore, this survey does not pose ethical or societal concerns.

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
