# OpenReview forum: "Implicit Reasoning in Large Language Models: A Comprehensive Survey"
_TMLR — Withdrawn by Authors_

### Review · Reviewer_sa3e · 2025-10-13

**Summary Of Contributions:**

This is a nice paper organizing the research-works within the topic of implicit-reasoning (i.e., reasoning that is not directly human-interpretable).

Strengths:
- in my opinion, is that this work proposes an organization of papers based on design/training-objectives and serves as a good introduction to this topic.
- The authors promise to maintain this project with new papers/benchmarks post-publication and this can be a good resource.

Weaknesses:
- While I enjoyed the presentation of the papers, the ideas contained within them and the manner in which papers are connected together, I  found that they paper itself does not have any qualitative/quantitative comparison between these methods in terms of empirical/runtime gains. In fact, this is showcased as one of the challenges ("Lack of Standardized Evaluation"). Beyond an introduction to papers in this area, I find that this paper does not offer new insights on its own.

**Audience:**

Yes

**Audience Explanation:**

Yes, this is an exciting area of research and the paper can become a key reference for new research-works in this subfield.

**Claims And Evidence:**

No

**Claims Explanation:**

The paper makes three claims regarding contributions and I do not agree with this claim:
> We conduct a parallel investigation into the evidence for implicit reasoning by synthesizing findings
from structural analyses, behavioral signatures, and representation-based analysis techniques, providing empirical grounding for the internal dynamics captured by our execution-centric taxonomy.

In my reading, this suggestes that the authors carry out their own independent analysis and comparison between the various systems discussed in this paper but in reality, the paper seems to only present a narrative based on research findings/insights of other papers. I request the authors to either expand the paper with some of your own analysis/findings (this will certainly make it more impactful), tone down this claim or justify this.

**Requested Changes:**

Please see above. In addition, I wanted to share this paper for your review/consideration in case you haven't seen this already:
> Jain, Adit, and Brendan Rappazzo. "Learning to Reason with Mixture of Tokens." arXiv preprint arXiv:2509.21482 (2025).

---

### Review · Reviewer_NTTo · 2025-10-15

**Summary Of Contributions:**

This paper reviews the literature on implicit reasoning in LLMs and provides a taxonomy to classify relevant works in the space. Specifically, it looks at three kinds of works, categorizing methods for implicit reasoning into a nested hierarchy, categorizing forms of evidence for implicit reasoning’s existence, and providing a summary of metrics and a list of commonly used benchmarks. Finally, the paper concludes by discussing limitations of implicit reasoning.

The nested hierarchy for implicit reasoning provided divides techniques into ones that (3.1) modify the tokens or hidden states the model processes, (3.2) add reasoning tokens to the tokens the model produces, and (3.3) ones that do not proceed purely autoregressively, instead running layers iteratively on certain states without the need to produce output tokens at each step.

The categorization of lines of evidence for implicit reasoning divide this evidence into (4.1) evidence found in analyses of individual layer activations of transformers, (4.2) end-to-end evaluations of the models’ behavior on certain tasks, and (4.3) detailed analyses of the activation states of transformers to attempt to identify specific computational patterns.

The paper then discusses benchmarks, starting with a discussion of metrics (5.1) for correctness, efficiency, perplexity, and probing accuracy; benchmarks. It then covers (5.2) benchmarks for general knowledge/commonsense reasoning, mathematical/programming tasks, language modeling, complex question-answer, and multimodal reasoning.

Finally, the paper discusses limitations within the current field of LLM implicit reasoning.

This paper appears to be quite comprehensive, covering an impressive array of work in the field of implicit reasoning. However, there are several significant flaws with both clarity and organization that are discussed in later sections of this review. Additionally, Section 5.2, one of the longest sections of this paper, appears to be a survey of datasets for LLM reasoning, which seems somewhat out of scope for this paper, which is about specifically LLM implicit reasoning. Finally, the three main sections (Sections 3, 4, and 5), are not directly tied together.

**Additional Comments:**

None. See above sections for all my comments.

**Audience:**

No

**Audience Explanation:**

I believe that the topic this paper is covering would be of interest to the TMLR audience; however, as written, the paper lacks the clarity to be accessible to someone who has not read a majority of the papers covered, and does not have a sufficient discussion of gaps in the literature to be of interest to those readers who are deeply familiar with the field already.

**Broader Impact Concerns:**

I do not believe that there are any broader impact concerns related to this work.

**Claims And Evidence:**

No

**Claims Explanation:**

As this paper is primarily focused on the creation of a novel taxonomy of implicit reasoning in LLMs, the primary implicit claim of this paper is that the provided taxonomy is the most natural one available; and that this taxonomy either can be used to choose which technique to use in a particular context, or can be used to identify potential gaps in the existing literature.

As such, a key concern with Section 3 is that 3.1.1, which involves injecting tokens into a model, is quite similar to 3.2; and there is no explicit discussion of why these should not be grouped together rather than being in separate sections. Additionally, within 3.1, while the top level division generally makes sense, between (3.1.1) techniques that add individual additional tokens, (3.1.2) techniques that attempt to refine the entire space of tokens being processed (e.g., by compression), and (3.1.3) techniques that model hidden states. However, some papers seem oddly placed. For example, what makes SoftCOT appear in 3.1.2 rather than 3.1.1? As described, it places tokens within the LLM’s embedding space, which sounds like adding additional tokens. This is not to say that these papers are misplaced, but rather that the current paper does not provide sufficient argument for their current placement.

Similarly, Sections 4.1 and 4.3 have clear semantic overlap: both discuss analyses of activations at intermediate layers in transformers. A clear explanation of why these are distinct would be helpful.

Section 5.1.1 has similar problems, with the metrics described having semantic overlap. See Requested Changes for more details.

Beyond these specific concerns, a broader issue with evaluating the taxonomy (particularly in Section 3) is the lack of clarity in descriptions of the text. These clarity issues (discussed more in the Requested Changes) include (1) a lack of precision surrounding what the specific processes of training are [e.g., whether a particular set of tokens are trained end-to-end, extracted via some kind of unsupervised approach, etc.; whether the underlying LLM is frozen or not, etc.] (2) what the actual artifacts introduced are [e.g., does “learned token” indicate a specific learned vector, or does it represent a function mapping inputs/hidden states to vectors, etc.] (3) how much test time compute is involved in each technique. Additionally, papers are described using the terminology of the authors of each individual paper, rather than being rephrased within a unified language presented alongside the taxonomy.

This lack of precision and unification not only means that it is difficult for a new reader to understand what each of these papers’ contribution is but also makes it difficult to evaluate the taxonomy, as it makes it more difficult to see similarities and differences between the various techniques.

**Requested Changes:**

Section 2.2 and Section 2.3 are entirely redundant with Section 2.1, showing the same equation and clarifying that the symbols representing the intermediate state either represent outputs or hidden tokens; information already discussed in Section 2.1. These sections should be removed.

Section 3 header (end of page 6): “The first paradigm, latent optimization (§3.1), directly manipulates internal representations to improve reasoning without emitting intermediate text. The second, signal-guided control (§3.2), leverages specially designed control signals to steer the model’s internal computation process.” These sentences do not clarify the distinction between these paradigms. “Manipulat[ing] internal representations” and “leverages specially designed control signals” are non-mutually-exclusive ideas, absent additional context. It is important to clarify this distinction, as discussed later.

Section 3.1 and Figure 3 are difficult to understand and are not connected enough to each other. I have split my discussion of this Section and Figure into discussions of each subsection.

In Panel 3a, it is unclear what the dashed arrows, orange arrow, or upward arrows indicate. The dashed arrow is labeled gradient flow, is the idea that gradients are used to train the “Select Concepts” module, that is a sparse autoencoder? If so, the label for the entire module on the right should be Sparse Autoencoder. The plural makes it seem as though there are multiple, are there? Additionally, the connection between the module on the right and the diagram on the left is highly unclear, is the module on the right representing the upward pointing arrows? Or is it representing one of the green boxes? What does “a function model” mean in this context?

Panel 3b is more clear, but it remains unclear when exactly this insertion and training is happening. Reading Sun 2025, it seems that the process described here is a general pretraining step, with the “learnable latent tokens” being learnt once specialized to a model and not to a prompt, but neither the figure nor the caption makes it clear that this is what is happening.

Similarly, for Figure 3c, “Discrete latent tokens via vector quantization” is not sufficient, as it is unclear what the source of these vectors is before they have been quantized, and the figure does not make it clear. The text descriptions corresponding to this Figure are also insufficiently precise, the description of CoCoMix does not describe whether the LLM whose states are being manipulated is being retrained whether “salient concepts via attribution scores” are being selected on a per-prompt basis or not (I assume they are). The description in the text of Gong is also too high level, describing the goal of the discrete latent codes without describing where these come from or how they are used in the model.

Figure 3d needs a caption describing what “Mode 1” through “Mode 4” are. Additional twists that need clarification are whether these modes are a novel taxonomy introduced in this paper or do they come from Sun 2025, what the colors mean, and what makes modes 3 and 4 different from each other.

Section 3.2 and Figure 4 have similar problems. “Specifically, CCoT employs a scorer module to select the subset of gold hidden states and generates compressed tokens to approximate and align these subsets” is not a sufficient description of the process, as it does not clarify what a “gold hidden state” is, or what it means to generate a token to approximate a subset of states. Once again, there is a lack of clarity as to what states are, in this context. Do they refer to the intermediate states of an LLM on a particular input, or to the latent random variable of the LLM arising from the LLM being run on elements drawn from the input distribution? In other words, is this algorithm producing static tokens once via some training mechanism, producing a static function once via some training mechanism that generates tokens given the input, or something more online? This continues into the description of the other approaches (HCoT and CODI). SynAdapt’s description also does not explain why it uniquely “enabl[es] iterative refinement of draft trajectories without autoregressive generation;” the connection between generating an entire trajectory and no autoregression needs to be clarified in the text; given that the difference stated earlier has to do with the fidelity of alignment, not efficiency with respect to autoregression.

Section 3.3: “By adaptive token routing networks, ITT selects critical tokens of inner thinking layers to allocate additional thinking steps for deeper thinking” is unclear. Does this mean that the tokens are selected and then layers are repeatedly run just on these tokens? Or are they used as controls to decide how many additional steps to allocate? The description of RELAY is also unclear, what does alignment mean in this context, is it a form of loss supervision or is it structural? Some details on the “proposed iteration-wise alignment mechanism” might be clarifying. The last sentence of the description also implies that the purpose here is to improve the accuracy of reasoning traces rather than efficiency, which is not among the stated purposes of implicit reasoning as described in Section 2.4.

Section 5.1.1: Accuracy and EM are identical, as described. It seems as if Exact Match Accuracy is being contrasted with Pass@k, and then separately with BLEU/ROUGE, as two separate relaxations, one distributional, and the other approximate at an individual sample level. This structure should be clarified in this section, and the possibility of Pass@k BLEU/ROGUE should be discussed (if only to say that this is an incoherent concept or is never used).

Sections 3, 4, and 5 need to be tied together more. The following is my suggestion, but it is only one possible way to accomplish this: perhaps some kind of matrix figure showing the connections between Sections 3 and 4 and Sections 4 and 5 might be beneficial, especially in light of one of the limitations in Section 6 being a lack of proper robust evaluation. I recognize that Tables 2 and 3 somewhat tie Section 3 to Section 5, but they only do so via benchmarks, and in a way that makes it difficult to notice patterns. Perhaps some kind of summary of e.g., section headers might be beneficial (e.g., a table where rows are Section 3.1, 3.2, 3.3 and columns are Sections 5.2.1-5.2.5, and each cell is a % of methods that use a given benchmark). Similar information should be provided for the other figures.


Minor changes:

In Figure 1, the Introduction and Preliminaries sections are not part of the taxonomy, so should not be included in this chart.

On page 17: “by analyzing the computation process of looped Transformers through directed acyclic graphs (DAGs). It is shown that looped Transformers can simulate DAGs layer by layer, enabling efficient parallel reasoning on deterministic tasks” is somewhat vague language and instead could be written “by analyzing the computation process of looped Transformers on deterministic parallel computation tasks. It is shown that looped Transformers can simulate the corresponding computational DAG layer by layer”.

On page 18: “explore latent multi-hop reasoning of LLMs using one-hop and two-hop prompts, evaluating whether the models internally recall the bridge entity and measuring the consistency of response outputs between one-hop and two-hop prompts”. Unless you have read the paper, “bridge entity” is not common enough terminology to know. This should be rewritten to be more elementary, and, ideally, incorporate some of the papers’ conclusions.

On page 18: Does Liu 2024b belong here and not in section 3.1.2? It seems to be a form of token compression?

On page 18: “quantify the capacities of reasoning leaps between individual tokens by designing non-English language responses to benchmark implicit reasoning of 18 LLMs, demonstrating that the models engage in genuine internal reasoning rather than relying solely on heuristics, especially for dense models” is too vague a summary.

On page 21: “In practice, probing metrics can be complemented with causal or intervention-based analyses to enhance interpretability”: is this a fact that you are introducing or does it relate to previous work? If it is novel, you should mention that this has not been done before, if it relates to previous work, that work should be cited.

Section 5.1.2: “Notably, some implicit approaches (e.g., token-wise depth adaptation or latent recurrence) introduce dynamic computation paths, making these metrics insufficient.” This applies to some but not all of these metrics, correct? If so, this should be clarified, if not, it should be clarified why adaptive computation would make metrics such as latency or amount of GPU utilization insufficient.

---

### Review · Reviewer_gEBa · 2025-10-30

**Summary Of Contributions:**

This is a survey paper that does a comprehensive review of implicit reasoning in LLMs, where reasoning occurs internally without generating explicit intermediate steps. The authors introduce a novel taxonomy organizing existing methods into three execution paradigms: latent optimization, signal-guided control, and layer-recurrent execution. The paper systematically reviews evidence supporting implicit reasoning capabilities in LLMs, examines evaluation methodologies, and identifies key challenges for future research.

**Key strengths**
1. This paper uses a well-structured taxonomy that shifts focus from representational forms to computational strategies
2. The paper has a relative comprehensive coverage of recent literature
3. The paper systematically analyzes evaluation practices across the field

**Key Weaknesses**
1. There is not enough discussion on the theoretical foundations underlying implicit reasoning
2. Some claims about implicit reasoning advantages are not sufficiently supported by empirical evidence

**Additional Comments:**

None

**Audience:**

Yes

**Audience Explanation:**

This survey addresses a timely and important topic in the LLM research community. As reasoning capabilities become increasingly central to LLM applications, understanding how these models perform internal computation without explicit step generation is quite relevant. The paper could be valuable to researchers working on reasoning, interpretability, and efficient inference in LLMs.

The relatively comprehensive coverage of evaluation practices and the identification of standardization issues could benefit researchers designing new reasoning methods or evaluation protocols. The challenges and future directions section provides a roadmap that could guide future research efforts in this new area.

**Broader Impact Concerns:**

The statement should be relatively adequate

**Claims And Evidence:**

No

**Claims Explanation:**

While the paper provides a quite extensive literature coverage, some key claims lack sufficient empirical support. e.g. The authors claims that implicit reasoning offers "better alignment with internal computation", but the claim is mostly supported by citing individual studies rather than analysis across multiple works. The taxonomy, while intuitive, is not validated through rigorous categorization criteria or inter-annotator agreement studies.

The paper's central claim that implicit reasoning represents a fundamentally different computational paradigm from explicit reasoning is not convincingly established. The evidence presented is largely observational, and the authors do not sufficiently address alternative explanations for the observed phenomena. For instance, the difference between "genuine multi-step reasoning" and "exploiting memorized knowledge" is acknowledged as problematic but not resolved.

Additionally, the performance comparisons between implicit and explicit methods are inconsistent across studies, and the authors do not provide sufficient analysis on when and why implicit reasoning might be preferred over explicit approaches.

**Requested Changes:**

**Critical**
1. A more rigorous theoretical framework explaining what constitutes "implicit reasoning" versus other forms of internal computation is needed. Clearer definitions and criteria that distinguish implicit reasoning from pattern matching or memorization are important for the paper. The current definition in sec. 2 is a bit too broad and could include many forms of pattern matching
2. The three paradigm taxonomy requires additional validation. The authors may consider performing a systematic categorization of existing methods with clear criteria and potentially inter-annotator agreement to demonstrate the taxonomy's utility and completeness
3. The claim of implicit reasoning's advantages needs more details. Although the authors acknoledges that "implicit methods often underperform explicit strategies such as Chain-of-Thought in terms of final accuracy", Systematic analysis of when implicit methods outperform explicit methods and under what conditions the trade-offs are favorable is important for the paper

**Non-critical**
1. The authors can consider including more discussion of the computational mechanisms that might enable implicit reasoning (maybe in sec. 4).
2. There could be more systematic discussion of when and why implicit reasoning fails, and perhaps give concrete examples and potential mitigation methods
3. It could be benificial to address the reproducibility challenges in implicit reasoning research and propose standards for future work (sec. 5).

---

### Note · Authors · 2025-11-13

**Comment:**

We would like to sincerely thank the Action Editor and all reviewers for their time, effort, and thoughtful feedback. We truly appreciate the careful reading of our work and the constructive comments provided throughout the review process.

Several strengths highlighted in the reviews, including the comprehensive and up-to-date coverage of recent literature, the well-structured and execution-centric taxonomy, the clarity of the overall organization, and the timeliness and relevance of the topic, are genuinely encouraging for us. We are also grateful for the reviewers’ recognition of the survey’s contributions in synthesizing multiple lines of evidence, summarizing evaluation practices, and offering a broad perspective on the emerging research landscape of implicit reasoning in LLMs. These positive comments mean a great deal to us, and we sincerely appreciate the reviewers’ acknowledgement of the effort we invested in this work.

At the same time, we also appreciate the reviewers’ observations regarding potential areas for improvement. The comments on enhancing the clarity of certain definitions, refining a few distinctions in the taxonomy, and polishing parts of the presentation are valuable and will help us further strengthen the next version of the survey. We will carefully consider these reasonable suggestions during our subsequent revisions.

After careful consideration, we have decided to withdraw the current submission. This will allow us to further polish the manuscript, integrate the constructive feedback we received, and prepare a more mature and improved version of the work.

We sincerely thank the Action Editor and all reviewers again for their dedication, insightful comments, and contributions to the future development of this paper.

**Withdrawal Confirmation:**

I have read and agree with the venue's withdrawal policy on behalf of myself and my co-authors.